# LOCALITY ALIGNMENT IMPROVES VISION-LANGUAGE MODELS

**Ian Covert, Tony Sun, James Zou,**[*] **Tatsunori Hashimoto**[*]
Stanford University
`{icovert, suntony, jamesz, thashim}@stanford.edu`

## ABSTRACT

Vision language models (VLMs) have seen growing adoption in recent years, but many still struggle with basic spatial reasoning errors. We hypothesize that this is due to VLMs adopting pre-trained vision backbones, specifically vision transformers (ViTs) trained with image-level supervision and minimal inductive biases. Such models may fail to encode the class contents at each position in the image, and our goal is to resolve this with a vision backbone that effectively captures both local and global image semantics. Our main insight is that we do not require new supervision to learn this capability – pre-trained models contain significant knowledge of local semantics that we can extract and use for scalable self-supervision. We propose a new efficient post-training stage for ViTs called *locality alignment* and a novel fine-tuning procedure called MaskEmbed that uses a masked reconstruction loss to learn semantic contributions for each image patch. We first evaluate locality alignment with a vision-only benchmark, finding that it improves a model's performance at patch-level semantic segmentation, especially for strong backbones trained with image-caption pairs (e.g., CLIP and SigLIP). We then train a series of VLMs with and without locality alignment, and show that locality-aligned backbones improve performance across a range of benchmarks, particularly ones that involve spatial understanding (e.g., RefCOCO, OCID-Ref, TallyQA, VSR, AI2D). Overall, we demonstrate that we can efficiently learn local semantic extraction via a locality alignment stage, and that this procedure benefits VLM training recipes that use off-the-shelf vision backbones.

## 1 INTRODUCTION

Auto-regressive VLMs are an exciting new type of model that emerged in the last couple years and has seen growing adoption (Alayrac et al., 2022). They are more flexible than previous multi-modal image-text models (Karpathy & Fei-Fei, 2015; Radford et al., 2021), leverage the reasoning abilities and open-ended nature of pre-trained language models (LMs) (Touvron et al., 2023; Jiang et al., 2023; Zheng et al., 2023), and have the potential to subsume many visual tasks that can be expressed in natural language with interwoven images (Lu et al., 2022; Chen et al., 2022a; Gupta et al., 2022).

However, current VLMs make a range of basic perceptual errors and struggle in particular with spatial understanding. Multiple recent works document such failures (Tong et al., 2024b; Rahmanzadehgervi et al., 2024), and weaknesses can be seen through benchmarks focused on object localization (Kazemzadeh et al., 2014; Wang et al., 2021), counting (Acharya et al., 2019) and relational question-answering (Liu et al., 2023a). Data limitations are part of the problem, because LMs might not fully exploit visual features without sufficient joint training. But we suspect that another issue is how these models leverage pre-trained vision backbones: the most popular current ViTs are trained with image-level supervision and minimal spatial inductive biases (e.g., CLIP and SigLIP; Radford et al. 2021; Zhai et al. 2023b), so they may fail to encode the necessary information for spatial reasoning. Ideally, we want a ViT whose representation is sufficient to predict class contents not only for the entire image but also for each region, which we refer to as *encoding local semantics*. Since most VLM training recipes either freeze or only partially train the ViT backbone (Liu et al., 2023c; Karamcheti et al., 2024; Laurençon et al., 2024; Lu et al., 2024; Bai et al., 2023), and because it may be difficult

---

[*]Equal advising.

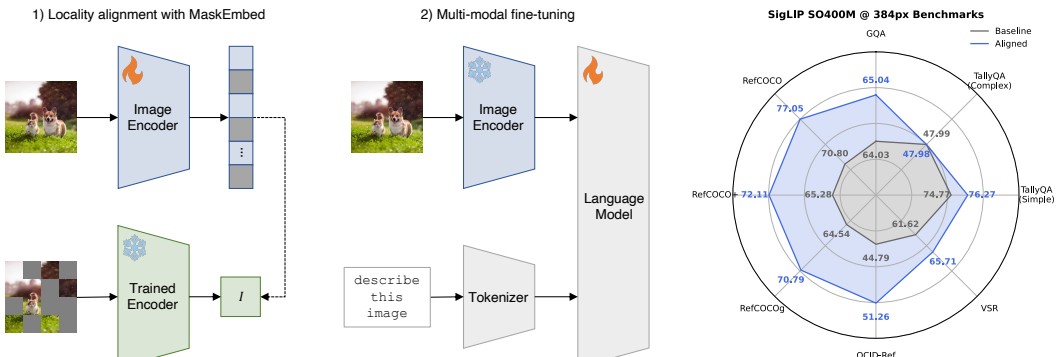

Figure 1: **VLM training pipeline with locality alignment.** Given a pre-trained vision backbone, we first perform a locality alignment stage using our MaskEmbed procedure (left), and then use the fine-tuned ViT to train a VLM (center). We find that doing so improves VLM performance in multiple benchmarks that involve spatial understanding (right).

to learn local semantics during joint fine-tuning without extensive multi-modal data, we reason that it would help to use a ViT that better captures these rich image details.

Our goal in this work is to train a vision backbone that matches the best existing models in global image understanding (Radford et al., 2021; Zhai et al., 2023b) but that also encodes local semantics. We reason that disentangling where semantics arise in an image provides necessary information for certain downstream tasks, and sacrifices nothing if local semantics collectively provide rich global image understanding. However, learning such a backbone is challenging due to limitations in current training approaches: for example, scalable objectives like CLIP offer only image-level supervision (Radford et al., 2021), semantic segmentation datasets contain relatively few images (Lin et al., 2014; Zhou et al., 2019; Gupta et al., 2019), and densely self-supervised methods like MAE and BEiT lack rich semantics (He et al., 2022; Bao et al., 2021).

Our main insight is that we do not require new supervision to learn this capability. We find that pre-trained models contain significant knowledge of local semantics that we can elicit by querying them with masked inputs: by examining counterfactual predictions under various masking patterns, we can analyze how the outputs change and infer semantics associated with each patch. We use this insight to design a fine-tuning procedure – we propose a *masked embedding self-consistency* (MaskEmbed) approach that uses masked patch embeddings to reconstruct masked views from the pre-trained model, and in doing so learns representations that capture localized image semantics.

Since we do not require training from scratch, we view this as a post-training stage for ViTs that we call *locality alignment* (Figure 1). The goal of this training stage is to take the set of concepts that an existing model is trained to recognize, and localize them by disentangling where they occur in an image. Our approach can be applied to any strong model trained with image-level supervision (e.g., CLIP, SigLIP, MoCo), leverages self-supervision instead of requiring costly human annotations, and has relatively low computational cost compared to pre-training. Our experiments focus on improving the performance of VLMs, but locality alignment may also prove useful for other downstream tasks.

To verify the effectiveness of locality alignment, we conduct both a vision-centric evaluation and a vision-language evaluation where we compare VLMs trained with different vision backbones. In our first set of experiments, we want to test whether locality-aligned ViTs encode what's where in an image, and we measure this via a simple probing benchmark: we cast existing semantic segmentation datasets as a patch-wise multi-label classification problem (e.g., MSCOCO; Lin et al. 2014) and find that locality alignment improves the performance of various backbones trained with image-level supervision, particularly language-supervised models like CLIP and SigLIP (Radford et al., 2021; Zhai et al., 2023b). Next, our main set of vision-language experiments compare a series of VLMs trained with and without locality alignment. We train our models using the recently released Prismatic library (Karamcheti et al., 2024) and with the strongest current ViT backbones, and we find that locality alignment improves performance across a range of benchmarks, particularly those that involve

spatial reasoning (e.g., RefCOCO, OCID-Ref, TallyQA, VSR, AI2D). Through these experiments, we find that the best models for VLMs are reliably improved by locality alignment.

To summarize, our **main contributions** in this work include:

- We introduce a locality alignment post-training stage for ViTs to recover local semantics from models that primarily encode global image information. Our MaskEmbed procedure leverages self-supervision to avoid requiring extra annotated data, is especially suitable for language-supervised models like CLIP and SigLIP, and requires minimal compute relative to pre-training (<1% of CLIP and SigLIP's pre-training compute in our experiments).

- Our vision-centric evaluation shows that locality alignment reliably enhances a model's ability to predict patch-level class contents. For various backbones trained with image-level supervision, we find that their locality-aligned counterparts improve at local feature extraction, with especially strong improvements for large and high-resolution models like CLIP ViT-L @ 336px and SigLIP SO400M @ 384px that are used in most current VLMs.

- Our vision-language evaluation shows that we can incorporate locality-aligned backbones and improve VLM performance across a range of benchmarks. We perform a series of controlled comparisons with a shared training recipe, and we observe improvements on multiple tasks including object localization, text understanding, counting and relational question-answering.

Overall, our findings reveal a gap between current pre-trained ViTs and the needs of open-ended VLMs for localized image semantics. Given the low cost and consistent improvements from MaskEmbed, our results suggest that locality alignment is a promising idea to incorporate into existing VLM recipes, and potentially for other downstream tasks that require spatial understanding.

## 2 RELATED WORK

**ViT pre-training.** There are many ways to pre-train ViTs, including strongly supervised approaches like image classification (Dosovitskiy et al., 2020), language-supervised objectives like CLIP and SigLIP (Radford et al., 2021; Yu et al., 2022; Zhai et al., 2023b; Tschannen et al., 2023), and various self-supervised tasks like BERT-style masked image modeling (Bao et al., 2021; He et al., 2022), augmentation-invariance (Chen et al., 2020b; Caron et al., 2021) and auto-regressive pixel generation (Chen et al., 2020a; El-Nouby et al., 2024). Pre-trained vision models are often adapted to downstream tasks, including semantic segmentation, object detection and depth estimation (Li et al., 2022b; Birkl et al., 2023; Kirillov et al., 2023), but training data for these tasks is typically scarce. Among these various training approaches, language-supervised models have proved most effective for VLMs in recent studies (Karamcheti et al., 2024; McKinzie et al., 2024; Tong et al., 2024a). Our work is motivated by a lack of training objectives with large-scale, dense and semantically rich supervision. We review existing pre-training approaches in more detail in Appendix A.

**ViT local feature extraction.** Several works have noted CLIP's lack of localized features in the context of downstream dense prediction tasks (Zhong et al., 2022; Rao et al., 2022; Xu et al., 2022; Wu et al., 2024). Other works have shown that ViTs learn to associate nearby patches (Dosovitskiy et al., 2020; Raghu et al., 2021; Jelassi et al., 2022), but this is distinct from encoding local semantics in their outputs. Some have proposed hybrid ViTs that reintroduce inductive biases from CNNs (Liu et al., 2021; Wu et al., 2021; d'Ascoli et al., 2021), but we improve the original ViT's local feature extraction without sacrificing expressive power. The works most closely related to ours are RegionCLIP (Zhong et al., 2022), CLIPSelf (Wu et al., 2024) and LocCa (Wan et al., 2024). RegionCLIP fine-tunes CLIP with synthetically labeled region-text pairs, which avoids human annotation but suffers from noisy caption matching. CLIPSelf fine-tunes CLIP to reconstruct features of random image sub-crops, which is similar to our approach but specifically intended for zero-shot semantic segmentation; this difference in goals leads to suboptimal localization under probing, as we show in Section 4. LocCa is trained to auto-regressively predict synthetic image captions from OWL-ViT (Minderer et al., 2022), which is itself a CLIP model fine-tuned on dense object annotations. Compared to LocCa, our approach requires significantly less compute, does not require any extra human annotations, and can be flexibly applied to any pre-trained model.[1]

---

[1] We are unable to compare to LocCa (Wan et al., 2024) due to a lack of public checkpoints.

**VLMs.** We focus on the class of auto-regressive vision-augmented LMs, which includes early examples like Flamingo, OFA, BLIP and Llava (Alayrac et al., 2022; Wang et al., 2022; Li et al., 2022a; Liu et al., 2023c), and current frontier models like Claude 3.5 Sonnet, GPT-4o and Gemini 1.5 (OpenAI; Anthropic; Reid et al., 2024). The most common approach to building such models is to combine a pre-trained ViT and a pre-trained LM (Bai et al., 2023; Lu et al., 2024; Beyer et al., 2024), which leverages strong capabilities learned from each modality. Several recent works investigate how to best integrate visual features (Laurençon et al., 2024; McKinzie et al., 2024; Karamcheti et al., 2024; Tong et al., 2024a). Most use high-resolution variants of CLIP or SigLIP for their vision backbone (Radford et al., 2021; Zhai et al., 2023b) and either freeze or only partially train the ViT alongside the LM, which makes it important for the initial ViT to capture local semantics.

**VLM perceptual failures.** VLMs are a diverse class of models with different interfaces and architectures, but many works have demonstrated perceptual errors across various types of multi-modal models (Thrush et al., 2022; Kamath et al., 2023; Yuksekgonul et al., 2023; Xu et al., 2024b). For the current generation of auto-regressive VLMs, perceptual flaws are apparent in benchmarks for counting, object localization, relational question-answering, object hallucination, and others like BlindTest (Rahmanzadehgervi et al., 2024) and MMMV (Tong et al., 2024b). Many of these tasks require spatial understanding, and we suspect that part of the problem is a failure to encode local image semantics. There are other ways to approach the issue, but an improved vision backbone composes with many of them: these include fusing features from multiple backbones (Karamcheti et al., 2024; Jain et al., 2024) or multiple image crops (Liu et al., 2024; Xu et al., 2024b), adding extra parameters for image processing (Tong et al., 2024a), and training with more data focused on spatial reasoning (Lu et al., 2022; Wang et al., 2023b; Peng et al., 2023; Xu et al., 2024a).

## 3 LOCALITY ALIGNMENT

Our goal is to train a vision backbone that encodes semantics both for the image as a whole and for each image region. Rather than training from scratch, we propose to address this in a post-training locality alignment stage. Our main insight, described in this section, is that pre-trained models offer a way to infer local semantics via masking. We show how to extract this information by querying the model with multiple masked images, and then how to make it more easily accessible by fine-tuning the model with self-supervision.

### 3.1 MASKING IS ALL YOU NEED

Consider a model trained to extract a rich global representation but no specific information for each image region, e.g., a CLIP image encoder (Radford et al., 2021). We want to use such a model to understand what's where in the image, and we propose to do so with masking. A model that accurately represents global image contents will change its output in response to input masking, and we can exploit this to probe a model under different masked views and understand each patch's contribution to the prediction. For example, comparing the output before and after masking a single patch provides information about that region's contents (Zeiler & Fergus, 2014).

We can build on this by masking multiple parts of the image and modeling the differences when each patch is masked. The simplest implementation is an additive approximation: if the model output is a vector, we can learn vectors of the same size for each patch and train them such that the partial summation approximates the masked output. Concretely, consider an input image $x$ represented as a set of $n$ patches $x = \{x_1, \ldots, x_n\}$, a binary mask $m \in \{0, 1\}^n$, and a masked image $m(x) = \{m_1 \cdot x_1, \ldots, m_n \cdot x_n\}$ where masked patches are set to the dataset mean. Given a pre-trained model $f(\cdot)$ with masked outputs $f(m(x)) \in \mathbb{R}^d$, we can write the patch embeddings as vectors $g_1, \ldots, g_n \in \mathbb{R}^d$ or as a matrix $g = [g_1, \ldots, g_n] \in \mathbb{R}^{n \times d}$, and we can train them such that $m^\top g \approx f(m(x))$ for a fixed image $x$ and all masks $m$.

This approach is a reasonable starting point, and it illustrates that pre-trained models contain latent knowledge of local semantics that can be extracted via masking. It also has a precedent in the literature: querying pre-trained models with masked images was one of the earliest approaches to zero-shot semantic segmentation (Xu et al., 2022), and this learning approach is the basis of certain interpretability methods (Jethani et al., 2021; Covert et al., 2022). However, we find that the additive approximation is limiting and not very effective in our experiments; this is because 1) patch semantics

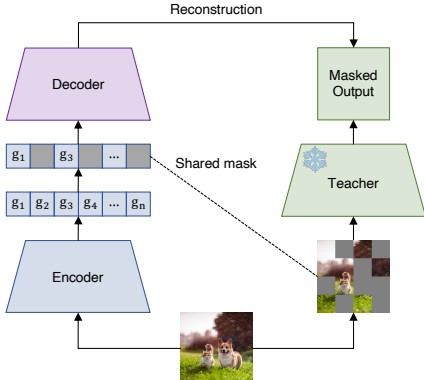

Figure 2: **MaskEmbed training diagram.** The encoder and decoder jointly reconstruct the pre-trained teacher's masked output, where patches are masked at the embedding layer for the encoder and at the input layer for the teacher.

aren't truly additive and the poor approximation causes us to lose information about each patch, 2) vector embeddings only allow us to reconstruct vector targets (e.g., the [CLS] token), which contain a small part of the pre-trained model's information about the image. Our main approach presented in the next section therefore generalizes this idea to learn richer patch embeddings.

## 3.2 PROPOSED APPROACH

We now introduce MaskEmbed, our fine-tuning procedure to enhance a model's local feature extraction abilities. Our basic idea is still to learn each patch's semantics by reconstructing masked views, but rather than doing so with an additive approximation we now use an expressive reconstruction function, and we obtain the patch embeddings by fine-tuning the pre-trained model.

We now let the patch embeddings be generated by a model $g_\theta(x) \in \mathbb{R}^{n \times d}$, which we refer to as an encoder and initialize with weights from the pre-trained ViT. We view the pre-trained model $f(\cdot)$ as a teacher whose masked views $f(m(x))$ are the reconstruction targets given the encoder's equivalently masked output $m(g_\theta(x)) \in \mathbb{R}^{n \times d}$, which we implement by setting masked embeddings to zero. We perform the reconstruction step using a transformer $h_\phi(\cdot)$ that we call a decoder, and whose predictions are denoted $h_\phi(m(g_\theta(x)))$. Importantly, the decoder can map to the teacher's output space regardless of its size, so we can adopt either the [CLS] token ($\mathbb{R}^d$) or an entire embedding layer ($\mathbb{R}^{n \times d}$) as the reconstruction target. To summarize, our model is trained with the following loss function in expectation over images $x$ and random masks $m$:

$$\min_{\theta, \phi} \mathcal{L}(\theta, \phi) = \mathbb{E}_{x,m}\left[\left\|h_\phi\big(m\left(g_\theta(x)\right)\big) - f\big(m(x)\big)\right\|^2\right]. \tag{1}$$

We call this procedure *masked embedding self-consistency*, or MaskEmbed for short, and Figure 2 shows a detailed training diagram. The pre-trained model weights are used to initialize the encoder and frozen teacher model, and the decoder is trained from scratch. The intuition behind this approach is that to minimize Equation (1), the encoder's output embeddings must represent semantics for each patch without leaking information from neighboring patches or the image as a whole. We expect the sequence of patch embeddings to collectively encode rich local and global information, which should be useful when training open-ended VLMs.

Compared to the simpler additive reconstruction approach (Section 3.1), MaskEmbed's use of an expressive decoder helps compress more information into each patch embedding. This also differentiates our approach from CLIPSelf (Wu et al., 2024), which adopts a related objective but aggregates CLIP's features by average-pooling within crop windows. We show the importance of this design choice in Section 4, where we also perform an ablation study to determine several hyperparameters for MaskEmbed. We remark that the main disadvantage of our approach is that our patch embeddings are less interpretable, because they lie in a different embedding space than the

pre-trained model's outputs; however, we reason that this is acceptable because our eventual use case is training a VLM that can learn how the entire representation encodes semantics.

### 3.3 TRAINING DATA

MaskEmbed is supervised by the pre-trained model's masked outputs, which means we can use any image dataset regardless of its annotations or lack thereof. Diverse data covering the pre-training distribution will help localize the broadest possible semantics, ideally including many types of objects, backgrounds, textures, facial features, etc. We use ImageNet-1k and ImageNet-21k (hereafter IN1k and IN21k) (Deng et al., 2009) for all our experiments, which are relatively diverse and contain 1.2M and 12.6M images in our training sets. A promising idea that we leave to future work is using larger web-scraped image datasets like those used for contrastive learning (Schuhmann et al., 2022; Xu et al., 2023; Gadre et al., 2023; Fang et al., 2023a), which are even more diverse and could help learn strong localized text features that are less prominent in ImageNet.

Related to training data, we note that our approach only works as intended if the pre-trained model makes meaningful predictions with masked inputs. This can be ensured by pre-training with randomly dropped patches, which is performed for some but not all of the models in our experiments (He et al., 2022; Bao et al., 2021; Peng et al., 2022; Fang et al., 2024). Training or fine-tuning with random masking is often suggested in the interpretability literature (Frye et al., 2020; Covert et al., 2021; Jain et al., 2022) because masked images are out-of-distribution if the model was not trained with masking, but we do not explore this direction and instead rely on the fact that ViTs empirically behave reasonably under masking (Naseer et al., 2021).[2]

## 4 VISION-CENTRIC EXPERIMENTS

For our experiments evaluating locality alignment, we aim to test whether MaskEmbed can successfully preserve an existing model's semantics while disentangling where they occur in an image. We initially want to do so without the complexity and computational cost of training a VLM, so we create a probing benchmark inspired by semantic segmentation. We first use this to determine several unspecified hyperparameters for MaskEmbed (e.g., the choice of reconstruction target), and then to compare a suite of pre-trained models to their locality-aligned counterparts.

### 4.1 PROBING BENCHMARK

A natural task to test whether a ViT encodes local image semantics is semantic segmentation (Long et al., 2015). However, this is a pixel-level classification problem, and the most performant approaches for ViTs require fully fine-tuning the backbone (Li et al., 2022c; Chen et al., 2022b; Fang et al., 2023b), sometimes with windowed self-attention (Li et al., 2022b). We want to test how well a ViT captures local semantics without task-specific fine-tuning, so we simplify the problem by casting it as a patch-level multi-label classification problem and keep the backbone frozen. Specifically, we create a small output head on top of the ViT's output embeddings, and we train it to predict the union of labels in each patch using a binary cross-entropy (BCE) loss. We implement this approach with MSCOCO (Lin et al., 2014), but we can also use other datasets like Ade20k (Zhou et al., 2019).

The performance on this patch-level task tests how well a model captures local semantics, and for a corresponding measure of global image semantics we also train output heads to predict the union of classes in an entire image; we refer to these tasks as *local probing* and *global probing* respectively, and we use macro-averaged recall as a performance metric that accounts for class imbalances in MSCOCO (Lin et al., 2014). We use two-layer transformer output heads unless otherwise specified, because this tests the information contained in the entire representation and is most similar to how a VLM uses the ViT output; Appendix B also shows results with other output heads.

---

[2]In Appendix B.4, we conduct initial experiments that suggest further gains if locality alignment is preceded by fine-tuning with randomly masked patches.

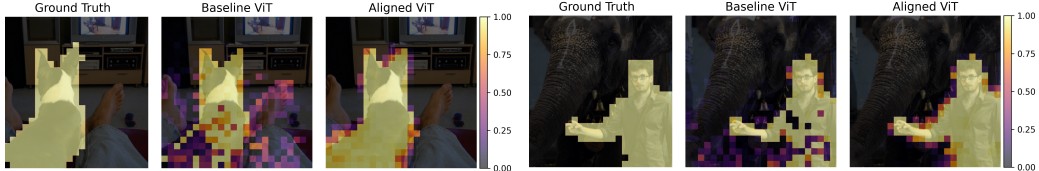

Figure 3: **Qualitative examples from probing benchmark.** We plot predictions for two images using CLIP ViT-L @ 336px before and after locality alignment. The original backbone fails to distinguish where certain objects occur in the image, but the aligned backbone corrects this.

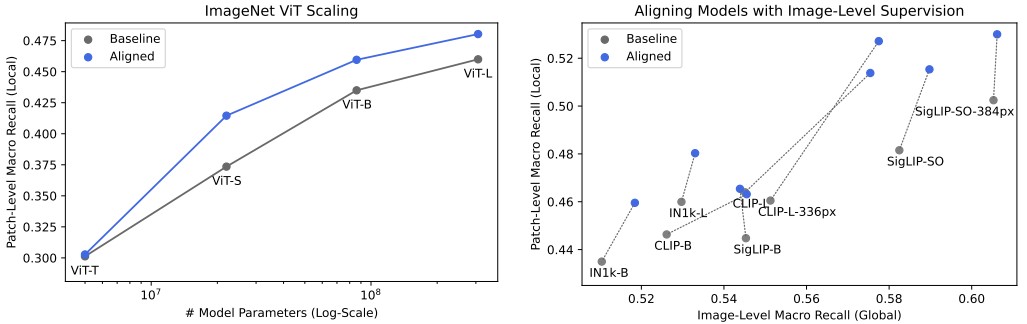

Figure 4: **Probing benchmark results.** We find that locality alignment with MaskEmbed improves IN1k classifiers across multiple model scales (left), and improves many models trained with language supervision (right). Interestingly, most models increase both their local and global probing accuracy.

## 4.2 ABLATING MASKEMBED DESIGN CHOICES

Our first usage of the probing benchmark is to explore several design choices for MaskEmbed. There are certain hyperparameters that we did not fully specify in Section 3.2, including the choice of reconstruction target and mask distribution, and we also want to test the importance of data augmentations, training duration and data diversity (IN1k vs. IN21k). We consider two pre-trained models for these experiments, IN1k ViT-B/16 and CLIP ViT-B/16 (Dosovitskiy et al., 2020; Radford et al., 2021), and we conduct a series of ablations to investigate these implementation choices.

We report the full results of our ablations in Appendix B, but we describe our main findings here that inform settings for our later runs. **Reconstruction target:** we observe that reconstructing the [CLS] token improves local probing performance, but not as much as reconstructing the entire embedding sequence from the second-to-last layer; this is expected, and we adopt this choice for the rest of our experiments. **Mask sampling:** we find that multiple mask distributions are effective, including the block masking approach from BEiT (Bao et al., 2021). We adopt an unstructured mask whose cardinality is sampled uniformly at random, and we additionally train with the complement of the mask and a mask that preserves all patches at each iteration.[3] **Data augmentations:** we observe that strong augmentations like Mixup, CutMix and AutoAugment are not necessary (Zhang et al., 2017; Yun et al., 2019; Cubuk et al., 2018), and we use a simple random crop for our main runs. **Decoder size:** performance is not very sensitive to the decoder size, so we adopt a simple two-layer transformer. **Training data:** we find that local probing performance improves within just 2 IN1k epochs, and that we can get strong improvements in under 50 epochs. We also find that training with the more diverse IN21k is important for CLIP ViT-B/16, which is pre-trained with more diverse data and can degrade when fine-tuned for too long with IN1k. For our remaining runs we train all models with IN21k for 5 epochs, which is equivalent to roughly 60k gradient steps with batch size 1024. Notably, this is less than 1% of pre-training cost for CLIP and SigLIP (Radford et al., 2021; Zhai et al., 2023b), so the marginal cost of locality alignment is low.

---

[3]In our notation this corresponds to $p(m) = 1/\binom{n}{|m|}(n+1)$, and at each step we calculate the reconstruction loss for three masks: $m \sim p(m)$, $1 - m$ and $\mathbf{1}$.

### 4.3 COMPARISON WITH PRE-TRAINED MODELS

We now perform experiments to verify that MaskEmbed improves local feature extraction for a range of pre-trained models. We consider ViTs trained with multiple forms of image-level supervision, including IN1k classifiers (Dosovitskiy et al., 2020), CLIP (Radford et al., 2021), SigLIP (Zhai et al., 2023b), other language-supervised models (OpenCLIP, DFN, EVA02; Cherti et al. 2023; Fang et al. 2023a; 2024) and MoCo v3 (Chen et al., 2021). Not all of these models are relevant for high-performance VLMs (Tong et al., 2024a), but we aim to test whether locality alignment works for any model pre-trained with image-level supervision. We use the settings determined in our ablation study, which include reconstructing the teacher's entire embedding sequence and training with IN21k for 5 epochs. Other details on our MaskEmbed hyperparameters are described in Appendix D.

Overall, we find that MaskEmbed reliably improves local probing performance for all these models, and in many cases even improves their global probing performance. Figure 4 (left) shows the local probing accuracy for IN1k models across different scales, where we find that performance improves for all models except the low-capacity ViT-T: locality alignment boosts the ViT-B's performance to roughly that of the next model scale, and provides a similar absolute improvement for the ViT-L. Next, Figure 4 (right) shows results for a range of models, including three CLIP and three SigLIP backbones, all of which improve substantially. Notably, the two strongest backbones for VLMs show clear improvements (CLIP ViT-L @ 336px and SigLIP SO400M @ 384px), suggesting that the challenge of learning local semantics is not solved merely with scale, but is significantly improved by locality alignment. Figure 3 shows qualitative examples from CLIP ViT-L @ 336px, demonstrating how MaskEmbed helps identify where each object occurs in the image. Appendix B shows results for the remaining models, all of which show similarly large improvements (OpenCLIP, DFN, EVA02, MoCo v3); we find that locality alignment can even improve probing performance for some densely supervised models, including BEiT and BEiTv2 (Bao et al., 2021; Peng et al., 2022). In addition, we corroborate these results by showing that locality alignment improves IN1k classification accuracy, which represents a more challenging global image understanding task (see Appendix C).

Table 1: **CLIPSelf comparison.** We compare MaskEmbed to CLIPSelf's crop-based objective using CLIP ViT-B. For fair comparison we include a version of MaskEmbed with averaged features instead of a transformer decoder, and a version that uses just one mask per batch rather than three. Results that are worse than the teacher are shown in red.

|  | # augs/batch | local | global |
|---|---|---|---|
| teacher |  | 44.63 | 52.61 |
| CLIPSelf | $1\times$ | 36.16 | 42.48 |
| MaskEmbed (avg embed) | $1\times$ | 40.97 | 47.68 |
| MaskEmbed | $1\times$ | 46.07 | 53.17 |
| MaskEmbed | $3\times$ | **46.32** | **54.55** |

Finally, we perform a comparison with CLIPSelf (Wu et al., 2024). This method uses a similar objective and reconstructs cropped views using cropped ViT features, but it reconstructs CLIP's `[CLS]` token by simply average-pooling embeddings within each crop window. We test this method in Table 1, where we find that it in fact degrades CLIP's probing performance. We suspect that the main issue is not crops but the use of a weak decoder (i.e., averaging features within the crop), and we verify that MaskEmbed also degrades performance when we use this approach to reconstruct the `[CLS]` token (averaging unmasked embeddings rather than passing them to a learned decoder). Our main version of MaskEmbed proves to be much more effective, although unlike CLIPSelf it does not preserve CLIP's zero-shot classification abilities.

## 5 VISION-LANGUAGE EXPERIMENTS

We now conduct our main experiments by training a series of VLMs with and without locality alignment, and checking for improvements in relevant benchmarks.

**Experimental setup.** We train VLMs using the Prismatic library and training recipe (Karamcheti et al., 2024). Images are turned into embedding sequences by the ViT (Liu et al., 2023c), projected

into the LM embedding space by an adapter module, concatenated with text token embeddings, and processed by the LM. We train in a single stage with the ViT frozen, following Karamcheti et al. (2024). Our experiments focus on two high-resolution vision backbones, CLIP ViT-L @ 336px and SigLIP SO400M @ 384px (Radford et al., 2021; Zhai et al., 2023b; Alabdulmohsin et al., 2023), which respectively have 306M and 400M parameters and represent images with 577 and 729 tokens. For our LM backbone we use Llama-2 7B Base (Touvron et al., 2023), which was found to outperform the instruction-tuned Vicuña 7B (Zheng et al., 2023) by Karamcheti et al. (2024).

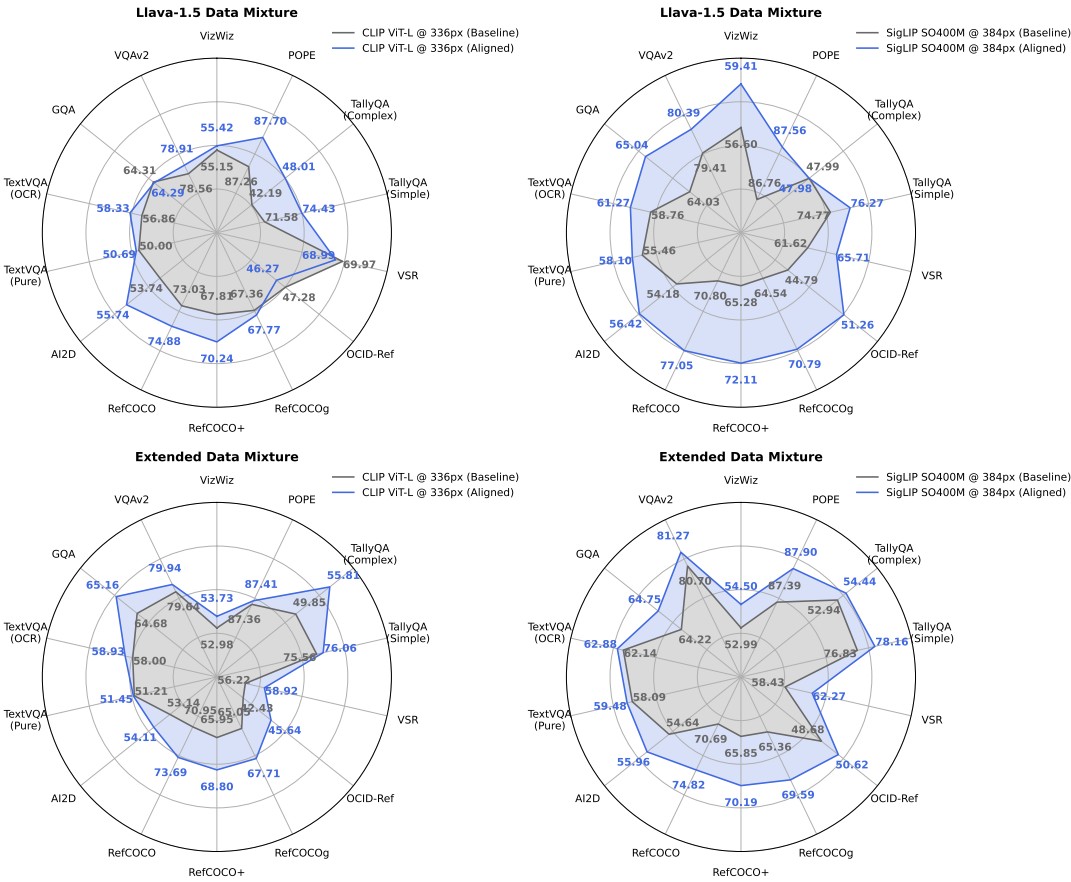

Figure 5: **VLM benchmarking.** We plot results across a suite of benchmarks and show controlled comparisons for CLIP (left) and SigLIP (right) with both the Llava-1.5 data mixture (top) and the extended data mixture (bottom). Overall, we achieve better performance in nearly all metrics with locality-aligned backbones. Between the two data mixtures, we find that the larger dataset does not have uniformly better performance and leads to different gains across text comprehension, chart understanding and localization tasks.

For our training dataset, we use the Llava-1.5 data mixture (Liu et al., 2024) that contains 665k examples, and which consists of synthetic instruction completions (Liu et al., 2023c), existing vision-language datasets (e.g., GQA, TextCaps; Hudson & Manning 2019; Sidorov et al. 2020) and a collection of language-only data (ShareGPT, 2023). We also experiment with an extended data mixture considered by Karamcheti et al. (2024), which adds LVIS-Instruct-4V (Wang et al., 2023a) and LRV-Instruct (Liu et al., 2023b) for an additional 570k examples. We provide more details on the training data in Appendix E, and all models are trained for two epochs.

**Evaluations.** We use a suite of standardized benchmarks considered by Karamcheti et al. (2024). Benchmarks that involve spatial understanding and fine-grained features include object localization (RefCOCO, OCID-Ref; Kazemzadeh et al. 2014; Wang et al. 2021), counting (TallyQA; Acharya et al. 2019), relational question-answering (VSR; Liu et al. 2023a), chart understanding (AI2D; Kembhavi et al. 2016) and text comprehension (TextVQA; Singh et al. 2019). We also show results

for holistic question-answering (VQAv2, VizWiz; Goyal et al. 2017; Bigham et al. 2010) and object hallucination (POPE; Li et al. 2023c), which are not as closely related to spatial understanding. We provide more details on our suite of benchmarks in Appendix E.

## 5.1 RESULTS

We show results in Figure 5 for the full suite of benchmarks. We plot metrics in radar charts for both CLIP and SigLIP, separating results based on the two data mixtures that we consider. Following prior work (Karamcheti et al., 2024), we scale each benchmark's y-axis based on the mean and standard deviation within our pool of models. We find that locality alignment is broadly useful and improves performance in most benchmarks, especially those mentioned above that involve spatial understanding. Notably, the generally stronger SigLIP SO400M @ 384px backbone (Tong et al., 2024a) has better performance in nearly all benchmarks using our approach.

For VLMs trained with standard backbones, we follow the exact training recipe from Karamcheti et al. (2024). But for those trained with locality-aligned backbones, we find that one small architecture change is necessary to achieve these performance improvements: rather than using the standard MLP vision-language adapter (Liu et al., 2024), we use the trained decoder module from MaskEmbed as an adapter (see Section 3.2). This unlocks robust performance improvements consistent with our probing results in Section 4.3, whereas using a MLP adapter applied to the fine-tuned embeddings slightly hurts performance (see ablations in Appendix E). We reason that this is because information is compressed into a space that is difficult to use compared to the text-aligned CLIP and SigLIP spaces, and that the decoder helps resolve this for the LM. Overall, the modified adapter adds negligible compute overhead and is a simple change to yield improved spatial understanding.

In Appendix E, we also show a comparison with an alternative approach to improving spatial understanding: fusing features from a second backbone, specifically DINOv2 (Oquab et al., 2023), following the implementation from Karamcheti et al. (2024). We find that both methods improve spatial understanding benchmarks like RefCOCO and TallyQA, with feature fusion in some cases leading to larger gains. However, we also observe that feature fusion can degrade the model in other ways that do not occur with locality alignment, including holistic question-answering (VizWiz) and text comprehension (TextVQA) – likely because text is not prominent in DINOv2's pre-training. We leave to future work a careful study of how to compose locality alignment with feature fusion, as well as other ideas like combining multi-crop features (Liu et al., 2024; Xu et al., 2024b), increasing image resolution (Bai et al., 2023) and utilizing prefix attention in the LM (Beyer et al., 2024).

## 6 DISCUSSION

Our main contributions in this work are proposing locality alignment as a post-training stage for ViTs, investigating a specific implementation with MaskEmbed, and demonstrating improvements in local feature extraction and VLM performance. We find that local feature extraction can be improved using only self-supervision, and that this is effective for many models trained with image-level objectives. Most notably, locality alignment boosts performance for VLMs that adopt the high-resolution CLIP and SigLIP backbones, which are widely used in recent works.

One limitation of our work is that we focus on a single VLM training approach – the Llava-style patches-as-tokens architecture (Liu et al., 2023c), and the specific Prismatic recipe of training in a single stage with the ViT frozen (Karamcheti et al., 2024). The benefits of locality alignment may change with end-to-end fine-tuning, but we did not explore this because it is unhelpful with our amount of multi-modal training data (Karamcheti et al., 2024). An important direction for future work is to test locality alignment in other VLM training approaches, with larger LMs, and to evaluate how it composes with other techniques that enhance visual features.

As other directions for future work, we speculate that locality alignment may yield larger gains when training for longer with more diverse data (e.g., DataComp; Gadre et al. 2023). Next, because we observe significant gains for large and high-resolution backbones, an exciting direction is to locality-align native-resolution ViTs (Dehghani et al., 2023b): these offer the potential to capture fine-grained details in large images, but due to their large token counts are at higher risk of mixing information across locations and losing local semantics. And finally, because MaskEmbed can be understood as leveraging synthetic data for large-scale dense supervision, it may be possible to

adapt our approach for end-to-end vision-language training and incorporate it into the pre-training data mixture for VLMs like Chameleon (Chameleon Team, 2024), and even vision encoder-free architectures like Fuyu (Bavishi et al., 2023).

## CODE

We provide repositories to reproduce each part of our results:

| | | |
|---|---|---|
| ⬤ | **Locality alignment** | `https://github.com/iancovert/locality-alignment/` |
| ⬤ | **Probing benchmark** | `https://github.com/iancovert/patch-seg/` |
| ⬤ | **VLM training** | `https://github.com/iancovert/prismatic-vlms/` |

## ACKNOWLEDGEMENTS

The authors thank Yann Dubois, Sheng Liu, Mert Yuksekgonul, Rahul Thapa, and other members of the Zou and Hashimoto labs for helpful discussions. We also thank the authors of the Prismatic library (Karamcheti et al., 2024), and Ross Wightman for creating and maintaining the `timm` repository.

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

# A    EXTENDED RELATED WORK

This section provides an extended discussion of related work, including our proposal's connections and differences with existing pre-training and distillation approaches.

**Other ViT pre-training methods.** The main text mentions a number of strongly supervised, language-supervised and self-supervised pre-training methods (see Section 2). We add to list this several more self-supervised methods including iBOT (Zhou et al., 2021), DINOv2 (Oquab et al., 2023), MoCo (Chen et al., 2021), CISSL/DISSL (Dubois et al., 2022), and pretext tasks like jigsaw puzzle solving (Noroozi & Favaro, 2016) and rotation prediction (Gidaris et al., 2018). Beyond these works that develop new objectives, other works explore combinations of multiple objectives (Mu et al., 2022; Kim et al., 2023; Dong et al., 2023; Chen et al., 2024), e.g., CLIP combined with SimCLR (Chen et al., 2020b) or CLIP combined with MAE (He et al., 2022). Other works combine pre-training with distillation from strong teacher models (Sameni et al., 2024). Compared to these works, locality alignment 1) relies on self-supervision instead of distilling from other strong models, 2) removes the need for augmenting pre-training objectives with secondary objectives to learn localized semantics.

**Knowledge distillation.** Knowledge distillation is a technique to train small models that imitate larger ones (Hinton et al., 2015) that works across many machine learning problems (Sanh et al., 2019; Taori et al., 2023). Deviating from the original motivation, some works have adopted versions of distillation for self-supervised learning (Caron et al., 2021; Baevski et al., 2022), and others use it for masked image modeling (Peng et al., 2022; Fang et al., 2023b) or to learn models that handle missing information for better interpretability (Frye et al., 2020; Jethani et al., 2021; Jain et al., 2022). MaskEmbed is a form of self-distillation because we reconstruct augmented teacher views, similar to works like Consistent Teaching (Beyer et al., 2022) and ReLabel (Yun et al., 2021). However, our use of masking at the embedding layer is a key difference that enables MaskEmbed to learn localized patch semantics.

**Comparison with other existing approaches.** In Table 2, we compare MaskEmbed to existing methods that use various combinations of masked prediction, dense supervision and knowledge distillation. MaskEmbed differs in its use of dual masking for both the student and teacher, because most methods only perform masking for the student model. Unlike other densely supervised methods, especially masked image modeling methods like MAE, BEiT and MaskFeat (He et al., 2022; Bao et al., 2021; Wei et al., 2022), we do not adopt single labels for each patch: MaskEmbed is the only method in Table 2 that supervises student predictions by decoding arbitrarily masked patch embeddings to reconstruct mask-dependent labels. Overall, MaskEmbed has important differences from prior works that enable learning rich localized semantics from a pre-trained teacher model.

Table 2: **Comparison to methods involving combinations of masked prediction, dense supervision and knowledge distillation.** [†]Unlike some previous works, we do not adopt single labels for each patch but instead let them change as a function of the mask. [‡]Unlike previous works, we perform student masking on patch embeddings rather than raw pixels.

| | Labels | Dense Supervision | Teacher Masking | Student Masking |
|---|---|---|---|---|
| MAE (He et al., 2022) | Raw pixels | ✓ | | ✓ |
| MaskFeat (Wei et al., 2022) | HOG features | ✓ | | ✓ |
| BEiT (Bao et al., 2021) | dVAE | ✓ | | ✓ |
| BEiTv2 (Peng et al., 2022) | Pre-trained model | ✓ | | ✓ |
| EVA (Fang et al., 2023b) | Pre-trained model | ✓ | | ✓ |
| data2vec (Baevski et al., 2022) | Momentum encoder | ✓ | | ✓ |
| FLIP (Li et al., 2023b) | Image captions | | | ✓ |
| CLIPA (Li et al., 2023a) | Image captions | | | ✓ |
| Masked Surrogate (Frye et al., 2020) | Pre-trained model | | | ✓ |
| Token Labeling (Jiang et al., 2021) | Pre-trained model | ✓ | | |
| MaskEmbed (Ours) | Pre-trained model | ✓[†] | ✓ | ✓[‡] |

## B    PROBING BENCHMARK DETAILS & ADDITIONAL RESULTS

**Output head.** All experiments with our probing benchmark use a frozen ViT and a trainable output head. The main text results use a transformer output head with two layers, learnable position embeddings, and the same model dimension and number of attention heads as the ViT backbone. We also include supplementary results in Figure 6 with linear and MLP output heads; the MLP output heads use one hidden layer of size 1024 and GELU activation.

**Hyperparameters.** All output heads are trained with the same approach using hyperparameters that we tuned for the non-aligned IN1k ViT-B/16 backbone (see Table 3). We use the training examples from MSCOCO with semantic segmentation masks (118k images) and report results using the validation set (5k images) (Lin et al., 2014). MSCOCO contains 183 total class labels split between *things* classes, *stuff* classes and the *unlabeled* class. We report macro-averaged recall for all results, as we found that with our multi-label classification setup the per-class 0-1 accuracy and AUROC are too high to show meaningful differences between models. All training runs are performed on a single NVIDIA H100 80GB GPU.

Table 3: **Probing benchmark hyperparameters.**

| Hyperparameter | Value |
| --- | --- |
| Epochs | 5 |
| Batch size | 32 |
| Weight decay | 0.01 |
| Augmentation | None |
| Gradient clipping | None |
| Optimizer | AdamW |
| $\beta_1, \beta_2$ | (0.9, 0.999) |
| Learning rate schedule | Linear warmup + cosine decay |
| Max learning rate | 1e-3 |
| Min learning rate | 1e-4 |
| Warmup steps | 500 |

### B.1    ABLATION STUDY

We report the full results from our MaskEmbed ablation study in Table 4. These results inform our settings for the reconstruction target, data augmentations, mask sampling approach, loss function, training dataset and training duration. Separately, we also found in our early experiments that varying the decoder depth and width did not lead to clear improvements; all our reported results therefore use a two-layer decoder with the same model dimension and number of attention heads as the pre-trained ViT. We describe each ablation parameter in detail below.

**Reconstruction target.** We consider three choices for the teacher reconstruction target: the `[CLS]` token from the last layer, the last layer's entire embedding sequence, and the second-to-last layer's embedding sequence. We find that the embedding sequences both work better than the `[CLS]` token, consistent with our intuition that all the tokens contain useful information. The last layer provides a larger improvement for global probing, and the second-to-last layer provides a large improvement for local probing. We use the second-to-last layer in our subsequent experiments.

**Data augmentation.** The minimum amount of data augmentation we can apply during MaskEmbed is a random crop and resize to the ViT's resolution, in this case $224 \times 224$ for both IN1k ViT-B and CLIP ViT-B. In addition to the random crop, we consider applying Mixup (Zhang et al., 2017), CutMix (Yun et al., 2019) and an AutoAugment recipe (Cubuk et al., 2018) as stronger augmentations. We find that Mixup and CutMix can help boost local probing performance but tend to hurt global probing performance. We opt to use the simple random crop in our remaining experiments, and we reason that strong augmentations are unnecessary because our masking leads to training each image with different reconstruction targets in each epoch.

**Mask sampling.** We consider several approaches to mask sampling. First, we use a block masking approach inspired by BEiT (Bao et al., 2021) that uncovers random rectangular regions until a desired

portion of the image is visible. Next, we consider a strategy that generates masks of roughly fixed size but without any structure: letting each position be revealed independently with the same probability (*Bernoulli*), similar to the MAE masking approach (He et al., 2022). Finally, we consider a *uniform* masking strategy that first samples the cardinality in $\{0, \ldots, n\}$ uniformly at random and then assigns the masked elements at random, which creates more variability in the portion of the image that is masked. We find that Bernoulli masking becomes more effective as we uncover larger parts of the image (75% vs. 25%), but that it does not lead to simultaneous gains in local and global probing. Our main experiments use the uniform approach with two modifications: in addition to the sampled mask $m$ we use its complement $1 - m$, and we also include the null mask that preserves all patches, which we find is helpful for global probing. These additions require extra compute, but crucially not from the encoder: the extra FLOPs are only incurred by the small decoder and the teacher model that does not require a backward pass for gradient computation, so this leads to just $1.66 \times$ the FLOPs of our base setting with a single mask (assuming a ViT-B backbone and a two-layer decoder).

**Loss function.** We compare several reformulations of our loss function presented in Equation (1). Our base setting is the MSE reconstruction loss calculated over all patches, and we find that this performs slightly better than either a cosine similarity loss or a $\ell_1$ loss that penalize deviations differently. We also compare to reconstructing only the masked patches or only the unmasked patches; while the latter performs slightly better for global probing, we find that the best approach for both local and global probing is to simply reconstruct all patches, which differs slightly from works like MAE and BEiT (He et al., 2022; Bao et al., 2021). We reason that this is because all the patch embeddings are non-trivial reconstruction targets in our setup, compared to MAE where unmasked patches can be reconstructed with the identity function.

**Training data and duration.** We compare training with IN1k and IN21k for different numbers of epochs. Our base setting is to train with IN1k for 25 epochs, and we find that performance improvements are mostly achieved even with minimal training (as few as 2 IN1k epochs). The best global probing performance is achieved in both cases with IN21k, whereas the best local probing performance is achieved with IN1k. One notable observation is that our performance does not always increase with longer training for CLIP ViT-B and can even degrade (see IN1k global probing); we suspect this is due to insufficient data diversity compared to the pre-training dataset. We choose to train with IN21k for 5 epochs in all our subsequent experiments.

|  | layer | local | global |
|---|---|---|---|
| teacher |  | 43.50 | 51.04 |
| [CLS] token | $L$ | 44.16 | 48.73 |
| embed seq | $L$ | 45.27 | 52.21 |
| embed seq | $L-1$ | 45.66 | 51.43 |

(a) **Reconstruction target.**

|  | local | global |
|---|---|---|
| in1k teacher | 43.50 | 51.04 |
| random crop | 45.66 | **51.43** |
| + auto-augment | 45.26 | 49.17 |
| + mixup | 45.72 | 51.34 |
| + cutmix | **46.59** | 48.60 |

(b) **Data augmentation.**

|  | FLOPs | local | global |
|---|---|---|---|
| in1k teacher |  | 43.50 | 51.04 |
| block | 1× | **45.66** | 50.29 |
| bernoulli 25 | 1× | 39.37 | 46.19 |
| bernoulli 50 | 1× | 43.55 | 46.86 |
| bernoulli 75 | 1× | 45.43 | 48.75 |
| uniform | 1× | 45.32 | 49.17 |
| + antithetical | 1.33× | 45.12 | 50.97 |
| + null mask | 1.66× | **45.66** | **51.43** |

(c) **Mask sampling.**

|  | local | global |
|---|---|---|
| in1k teacher | 43.50 | 51.04 |
| cosine | 45.55 | 51.37 |
| $\ell_1$ | 45.26 | 51.10 |
| mse | **45.66** | 51.43 |
| mse (masked) | 42.48 | 45.39 |
| mse (unmasked) | 45.00 | **51.67** |

(d) **Loss function.**

| dataset | epochs | steps | local | global |
|---|---|---|---|---|
| in1k teacher |  |  | 43.50 | 51.04 |
| in1k | 2 | 0.1× | 45.56 | 50.22 |
| in1k | 10 | 0.4× | 45.54 | 51.40 |
| in21k | 1 | 0.4× | 45.84 | 51.60 |
| in1k | 25 | 1× | 45.66 | 51.43 |
| in1k | 50 | 2× | 45.66 | 51.30 |
| in21k | 5 | 2× | 45.74 | **51.63** |
| in1k | 100 | 4× | **46.06** | 50.71 |
| in21k | 10 | 4× | 45.80 | 51.46 |

(e) **IN1k ViT-B/16 training data.**

| dataset | epochs | steps | local | global |
|---|---|---|---|---|
| clip teacher |  |  | 44.63 | 52.61 |
| in1k | 2 | 0.1× | 45.60 | 52.84 |
| in1k | 10 | 0.4× | 46.02 | 51.86 |
| in21k | 1 | 0.4× | 46.58 | 53.61 |
| in1k | 25 | 1× | **46.70** | 51.96 |
| in1k | 50 | 2× | 46.55 | 50.91 |
| in21k | 5 | 2× | 46.32 | **54.55** |
| in1k | 100 | 4× | 46.62 | 49.12 |
| in21k | 10 | 4× | 46.56 | 54.18 |

(f) **CLIP ViT-B/16 training data.**

Table 4: **MaskEmbed ablation study.** We ablate several task design choices using our probing benchmark, including the teacher reconstruction target, data augmentations applied on top of masking, the mask sampling approach, loss function, and the training data for two pre-trained models (IN1k ViT-B/16 and CLIP ViT-B/16). We report the local and global probing performance for all runs. The teacher model results are written in gray, our default settings are highlighted in gray, and the best results are **bolded**.

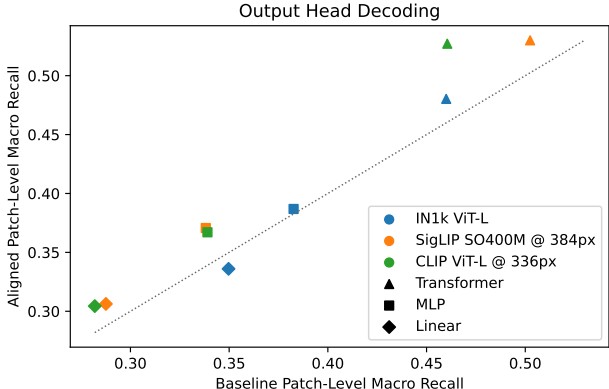

Figure 6: **Local probing performance with multiple output heads.** We show the improvement in local probing for three models when training three different output heads (transformer, MLP and linear).

## B.2 ADDITIONAL RESULTS

We now provide additional results from our probing experiments. First, Figure 6 shows results for three large models trained with three different output heads: IN1k ViT-L, CLIP ViT-L @ 336px, SigLIP SO400M @ 384px, and with transformer, MLP and linear output heads. We find that locality alignment improves performance not only with the transformer output head, but also with the other options (except for IN1k ViT-L with linear head). The transformer output head is the most relevant setting, but these results show that we successfully compress more relevant semantics for each patch into the corresponding embeddings and not just into the representation as a whole. However, it is notable that a large gap remains between the transformer output head and the others even after locality alignment; this shows that the embedding sequence learned by MaskEmbed is far more informative about a patch than the single corresponding patch embedding.

Next, Figure 7 examines one model to understand how our improvements are distributed across classes in MSCOCO (CLIP ViT-L @ 336px). We observe that our local probing performance improves roughly uniformly across all classes, with a few outliers. We also plot the top 10 most improved classes for both *things* and *stuff*; qualitatively, it appears that the most improved *things* classes are objects that could often be small in an image (e.g., cup, bottle, wine glass, scissors), which suggests that locality alignment may help better detect and localize non-dominant objects in an image.

Next, we test this by stratifying our improvements across object sizes. We group objects into 10 bins representing the portion of the image they occupy, and we re-compute the local probing performance within each bin. Figure 8 shows that we improve probing performance for objects of all sizes, but that locality alignment helps most for smaller objects. Again, this suggests that locality alignment can help better detect and localize non-dominant objects in images.

Next, we examine the probing performance across a suite of pre-trained models *without locality alignment*. Our goal is to better understand how well these models naturally encode local semantics, e.g., due to inductive bias in the ViT architecture. In Figure 9 (left), we plot the local and global probing accuracy for ViT-B models trained with a diverse set of pre-training objectives, including language supervision (CLIP, SigLIP, OpenCLIP, DFN, EVA02), self-supervision (MAE, DINO, DINOv2) and masked image modeling from pre-trained features (BEiT, BEiTv2).

It can be difficult to interpret absolute performance numbers in our benchmark, but we find that the comparative performance between models is informative. For example, we observe that local and global probing performance increase in tandem following a roughly linear trend (Figure 9). This suggests a notion of *relative locality* that describes how well a model performs at local probing given its performance at global probing, or simply how much it deviates from the empirical trendline. We note that certain models trained with dense self-supervision, including MAE and DINOv2, lie far above the empirical trendline. In contrast, models trained with image-level supervision sometimes lie

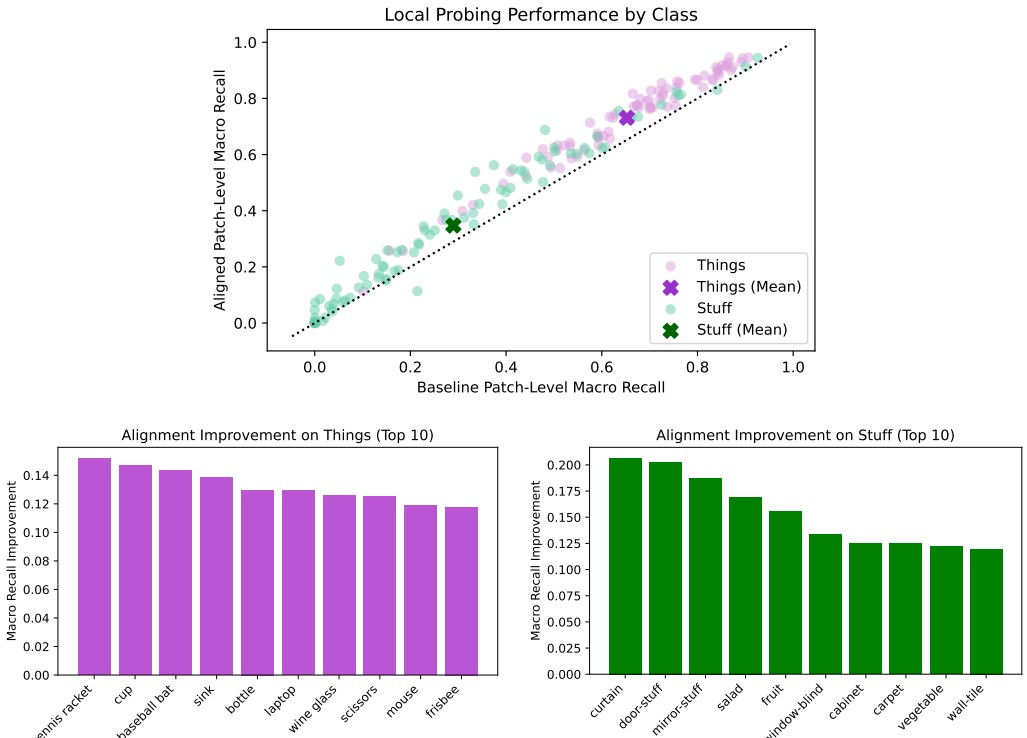

Figure 7: **Local probing improvements by class.** Results are shown for CLIP ViT-L @ 336px. We show the improvement for all classes (top), and we plot the top 10 most improved classes among both *things* (bottom left) and *stuff* (bottom right).

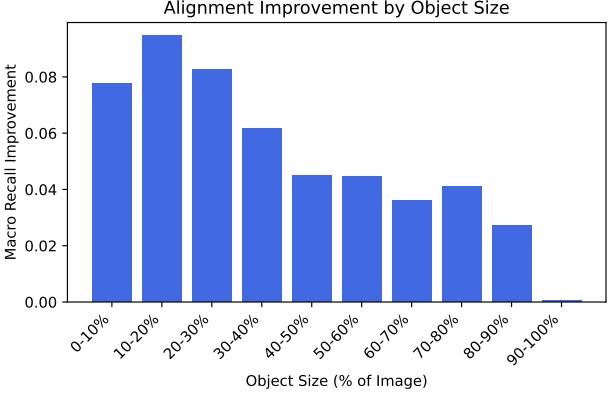

Figure 8: **Stratifying local probing improvements by object size.** Results are shown for CLIP ViT-L @ 336px.

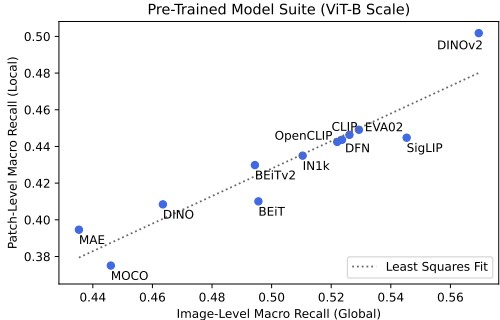 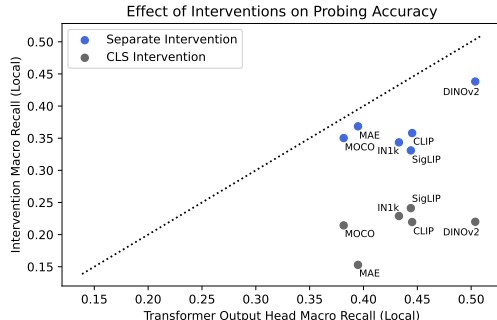

Figure 9: **Probing results for suite of pre-trained models.** We compare the local and global probing performance across a diverse set of models (left), and compare the local probing performance before and after applying interventions to remove spatial information from the ViT output (right).

Table 5: **Complete local probing results.** Results are separated by image-level supervision and various forms of dense supervision. Metrics that did not improve are highlighted in gray.

|  | Baseline | | Aligned | | Difference | |
|---|---|---|---|---|---|---|
|  | local | global | local | global | local | global |
| IN1k ViT-T | 30.13 | 41.26 | 30.28 | 40.89 | 0.15 | -0.36 |
| IN1k ViT-S | 37.35 | 46.37 | 41.46 | 46.20 | 4.10 | -0.17 |
| IN1k ViT-B | 43.50 | 51.04 | 45.96 | 51.84 | 2.46 | 0.80 |
| IN1k ViT-L | 46.00 | 52.97 | 48.03 | 53.30 | 2.03 | 0.33 |
| MoCo ViT-B | 37.50 | 44.60 | 40.38 | 45.29 | 2.88 | 0.69 |
| CLIP ViT-B | 44.63 | 52.61 | 46.32 | 54.55 | 1.68 | 1.94 |
| CLIP ViT-L | 46.40 | 54.51 | 51.38 | 57.54 | 4.99 | 3.03 |
| CLIP ViT-L @ 336px | 46.05 | 55.13 | 52.71 | 57.75 | 6.66 | 2.62 |
| SigLIP ViT-B | 44.48 | 54.53 | 46.54 | 54.39 | 2.06 | -0.14 |
| SigLIP SO400M | 48.15 | 58.25 | 51.54 | 58.98 | 3.38 | 0.73 |
| SigLIP SO400M @ 384px | 50.25 | 60.53 | 53.00 | 60.62 | 2.75 | 0.09 |
| OpenCLIP ViT-B | 44.25 | 52.20 | 45.17 | 52.62 | 0.92 | 0.42 |
| EVA02 ViT-B | 44.91 | 52.93 | 49.21 | 51.47 | 4.30 | -1.46 |
| DFN ViT-B | 44.36 | 52.36 | 45.67 | 53.72 | 1.31 | 1.36 |
| MAE ViT-B | 39.46 | 43.53 | 37.80 | 42.33 | -1.66 | -1.20 |
| BEiT ViT-B | 41.01 | 49.56 | 43.13 | 49.90 | 2.13 | 0.35 |
| BEiTv2 ViT-B | 42.98 | 49.44 | 46.60 | 53.58 | 3.62 | 4.14 |
| DINO ViT-B | 40.84 | 46.35 | 40.18 | 46.32 | -0.67 | -0.03 |
| DINOv2 ViT-B | 50.18 | 56.95 | 50.79 | 55.64 | 0.61 | -1.31 |

far below the line (MoCO v3, SigLIP); this indicates relatively poor local feature extraction and is a sign that locality alignment may be effective. Locality alignment is an intervention that can shift a model upwards and improve its relative locality.

Next, we consider what these results imply about how well ViTs naturally encode local semantics. Our work is motivated by the intuition that they may not, due to pre-training objectives that do not encourage it and a lack of inductive biases in the architecture, but in reality these models do not fail outright at the probing task. To emphasize this, we experiment with two interventions applied the transformer output head: 1) we restrict it to only have access to the `[CLS]` token (or the average embedding for models that do not use one), and 2) we anonymize the ViT's output embeddings by removing their learned positional embeddings and placing them in separate token positions from the predictions. Figure 9 (right) shows the probing performance before and after these interventions. It is clear that performance degrades due to these interventions, especially the first, suggesting that the ViT output does not collapse into a global representation containing no information about each patch's class contents. This is clear evidence that the patch embeddings provide useful information that significantly improves probing performance, even for models where these are not explicitly trained

(e.g., CLIP, IN1k). However, they generally do not perfectly capture local semantics and in many cases benefit from locality alignment.

Finally, Table 5 shows the results of running MaskEmbed on our full suite of pre-trained models. We observe that locality alignment improves local probing performance for all models trained with image-level supervision, and in most cases it also improves their global probing performance. The results are mixed for models trained with dense supervision: MAE, DINO and DINOv2 barely benefit from locality alignment (He et al., 2022; Caron et al., 2021; Oquab et al., 2023), and although BEiT and BEiTv2 do (Bao et al., 2021; Peng et al., 2022) this could be because we use checkpoints that are fine-tuned for IN1k classification.[4] We also note that results between different models are sometimes not comparable due to differences in resolution and patch size. Surprisingly, DINOv2 is the best-performing model overall despite being a relatively weak backbone for VLMs (Karamcheti et al., 2024; Tong et al., 2024a); we interpret this to mean that DINOv2 is exceptionally good at detecting and localizing the set of classes in MSCOCO, which are relatively narrow and perhaps not indicative of the diverse images handled by VLMs.

### B.3 CLIPSELF COMPARISON

We now describe our comparison with CLIPSelf (Wu et al., 2024) in more detail. We implemented a simple version of CLIPSelf where crops are aligned with the ViT's patch grid: we use CLIP ViT-B/16 (Radford et al., 2021), which operates on a grid of $14 \times 14 = 196$ patches, and for consistency with Wu et al. (2024) we sample crops containing between 3-14 patches on each side. The cropped image is then upsampled to $224 \times 224$ for the teacher model, which deviates slightly from the choice to pad in Wu et al. (2024). The student ViT's patch features are average-pooled within the crop window to reconstruct the teacher's `[CLS]` token, and we train the model with cosine similarity loss as in the original work. We sample one crop per image at each gradient step, and for a fair comparison we also run a version of MaskEmbed that uses just one mask per gradient step. When running our version of MaskEmbed that performs reconstruction via average-pooling, we use the block masking strategy (Bao et al., 2021) to avoid masks that contain no image patches. Unlike in the original CLIPSelf work we do not increase the student's resolution during training, which is a step that we also did not apply with MaskEmbed.

Figure 10 illustrates the masking and cropping operations involved in MaskEmbed and CLIPSelf. Both augmentations can meaningfully change the teacher's output depending on what contents are removed. Our results in Table 1 suggest that the main reason for CLIPSelf's poor performance is not the use of crops instead of masks, but the choice to reconstruct the teacher's `[CLS]` token by average-pooling features within each crop window. We speculate that a version of CLIPSelf that adopts a transformer decoder would be significantly more effective, but we leave this exploration to future work.

---

[4]We use checkpoints available on `timm` at `https://github.com/huggingface/pytorch-image-models`.

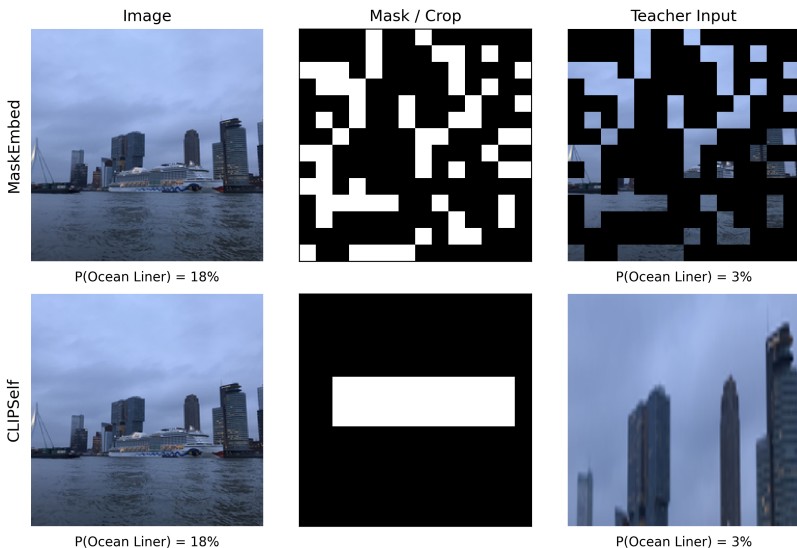

Figure 10: **Image transformations for MaskEmbed and CLIPSelf.** We show the original image, the randomly sampled image augmentation for each method (either a mask or crop), and the modified image seen by the teacher model. We annotate each image with class probabilities generated by IN1k ViT-B/16 to show that both augmentations can meaningfully change the teacher's output.

## B.4 Masked image modeling comparisons

In Section 3.3, we briefly discuss how locality alignment only works when the teacher model appropriately handles masked images, and how this can be encouraged by fine-tuning with random masking (Frye et al., 2020; Covert et al., 2021; Jain et al., 2022) before using locality alignment. We now explore the impact of including this additional fine-tuning stage.

In particular, we explore fine-tuning the ViT network with randomly masked input patches, and an objective of reconstructing embeddings from a frozen version of itself that processes entire images. Intuitively, this teaches the model to predict the semantics of the entire image by making its best guess for the missing patch contents, which is ideal behavior for the teacher model in locality alignment. We remark that this is similar to masked image modeling methods like MAE (He et al., 2022), but performing this task with rich features is known to work better than with raw pixels as the reconstruction target (Wei et al., 2022), and is the basis of recent masked image modeling works like BEiT v2 and EVA.

In Table 6, we test using locality alignment in different combinations with masked image modeling. Similar to locality alignment, we train for 5 epochs with IN21k and set other hyperparameters identically. We find that applying either fine-tuning approach to the original CLIP ViT-B backbone improves local and global probing, but that the gains are significantly higher for locality alignment. The gains on local probing are further improved if we perform masked image modeling followed by locality alignment, as predicted, which suggests that masked image modeling is a powerful precursor for locality alignment. Note that it could in principle be up-streamed into pre-training by simply dropping random patches, similar to FLIP (Li et al., 2023b). The best overall local probing performance is achieved by applying a subsequent round of masked image modeling fine-tuning, but this significantly degrades global probing. Meanwhile, the best overall global probing performance is achieved by applying locality alignment directly on the original CLIP ViT-B. For a relatively simple procedure that performs well at both probing tasks, a single round of masked image modeling followed by locality alignment is competitive at both evaluations, but we did not use this in any other experiments.

Table 6: **Combining masked image modeling with locality alignment.** We compare local and global probing performance for CLIP ViT-B models with different sequences of masked image modeling (MIM) and locality alignment. We find the the local probing performance can be significantly improved by performing locality alignment after an initial masked image modeling stage.

| Model | Teacher | local | global |
|---|---|---|---|
| CLIP ViT-B | – | 44.63 | 52.61 |
| CLIP ViT-B (MIM) | CLIP ViT-B | 45.80 | 52.98 |
| CLIP ViT-B (Align) | CLIP ViT-B | 46.32 | **54.55** |
| CLIP ViT-B (MIM/Align) | CLIP ViT-B (MIM) | 47.30 | 53.63 |
| CLIP ViT-B (MIM/Align/MIM) | CLIP ViT-B (MIM/Align) | **48.54** | 51.05 |
| CLIP ViT-B (MIM/Align/MIM/Align) | CLIP ViT-B (MIM/Align/MIM) | 47.80 | 53.34 |

## C  IMAGENET CLASSIFICATION

Our vision-centric experiments in Section 4 rely on the probing benchmark (see Appendix B), which assesses both local and global feature extraction but with a relatively narrow set of classes. To further test global semantic understanding and verify that it does not degrade with locality alignment, we also consider IN1k classification. We adopt an end-to-end fine-tuning setup with hyperparameters similar to those in OpenCLIP (Cherti et al., 2023), and we focus on CLIP backbones only for simplicity (Radford et al., 2021). One difference in our setup is that we replace the standard linear head for a transformer output head, because locality alignment teaches the model to output relevant semantics in all the output embeddings rather than just the [CLS] token.

The results are shown in Table 7, and show that the classification accuracy does not degrade but instead improves after locality alignment, across three ViT architecture variants. These results echo those in Table 5 for global probing, but in this case with end-to-end fine-tuning rather than a frozen backbone, and also a more challenging classification task. We attribute the improved performance to 1) the use of a full mask in MaskEmbed that leads to preserving global image understanding, and 2) an overall more challenging task that leads to stronger and more steerable internal features.

Table 7: **IN1k classification accuracy.** For each model, we perform end-to-end fine-tuning for 50 epochs before and after locality alignment, and we report the top-1 accuracy.

| Model | Baseline | Aligned |
|---|---|---|
| CLIP ViT-B | 82.6 | 83.1 |
| CLIP ViT-L | 85.9 | 86.3 |
| CLIP ViT-L @ 336px | 86.4 | 87.0 |

## D  MASKEMBED TRAINING DETAILS

We use this section to provide more details on our MaskEmbed implementation.

**Teacher model.** The teacher ViT is initialized from the pre-trained model weights and not updated during training. Its inputs are masked images, where masking is applied by setting masked patches to the image mean (or zero when images are normalized). Its output can be set in multiple ways, but we find that an entire layer's embedding sequence works best.

**Encoder.** The encoder ViT is initialized from the pre-trained model weights and updated throughout training. Its input is an unmasked image, and its output is a sequence of patch embeddings that go through an additional linear output head. We experimented with re-initializing the final transformer block because these parameters are typically pre-trained only to pass information to the [CLS] token (Dosovitskiy et al., 2020; Radford et al., 2021), but this did not improve performance.

**Decoder.** The decoder is a shallow transformer trained from random initialization, and we use LayerScale to ease its optimization (Touvron et al., 2021). Its input is a masked sequence of patch embeddings, and its output is a reconstruction of the masked teacher view. We extract the first entry from the output when reconstructing the [CLS] token, and we otherwise use the output at every position. We use learned position embeddings, omit the standard layer norm after adding position embeddings, and put the final output through a linear layer.

**Prefix token handling.** Most pre-trained models that we consider use a [CLS] token or other prefix tokens; our DINOv2 backbone uses extra register tokens (Darcet et al., 2023). For these models, it is unclear what role the prefix tokens should play in the reconstruction, because our goal is to compress semantics into the patch embeddings. We choose to mask prefix tokens at the decoder's input layer, but we keep them as part of the reconstruction objective.

**Training instability.** We encountered training instabilities in certain experiments, specifically a slow loss divergence that occurs partway through training. This type of instability has been reported in the literature with ViTs, with some works attributing it to saturation of the attention logits resulting in one-hot softmaxes (Zhai et al., 2023a); empirically, we were able to verify that diverged runs had a long tail of large attention logits. One common fix, QK-norm (Dehghani et al., 2023a; Chameleon Team,

2024), cannot be applied here because we fine-tune models that were pre-trained without QK-norm. We therefore use another approach that can be applied with a pre-trained model: logit soft-capping, where we use a `tanh` activation to constrain attention logits within a fixed range (Gemma Team et al., 2024). We adopt this approach in most of our MaskEmbed runs, including all runs that were used for training VLMs. We also had some success with increasing AdamW's $\epsilon$ parameter and increasing the weight decay to 0.1, but these sometimes led to slower optimization.

**Training data.** We experiment with running MaskEmbed using two datasets, IN1k and IN21k (Deng et al., 2009). We use the standard train and validation splits for IN1k, and we follow the pre-processing guidelines from Ridnik et al. (2021) for IN21k and create a validation set using sufficiently prominent classes.

**Hyperparameters.** We report hyperparameters for our main MaskEmbed runs in Table 8. All models are trained with AdamW (Loshchilov & Hutter, 2017), slightly lower $\beta_2$ than the default value, moderate weight decay, minimal augmentations, gradient clipping, cosine learning rate schedule, and batch size 1024. All MaskEmbed runs are performed on a single node with 4 NVIDIA A100 SXM4 80GB GPUs.

Table 8: **MaskEmbed hyperparameters.**

| Hyperparameter | Model scale | |
| --- | --- | --- |
| | ViT-T / ViT-S / ViT-B | ViT-L / ViT-SO400M |
| Global batch size | 1024 | 1024 |
| Weight decay | 0.01 | 0.01 |
| Gradient clipping | 1.0 | 1.0 |
| Optimizer | AdamW | AdamW |
| $\beta_1, \beta_2$ | (0.9, 0.95) | (0.9, 0.95) |
| Learning rate schedule | Cosine decay | Cosine decay |
| Max learning rate | 3e-4 | 2e-4 |
| Min learning rate | 3e-5 | 2e-5 |
| Augmentations | Random crop | Random crop |

## D.1 ADDITIONAL PERSPECTIVES

This section discusses some additional perspectives and observations about MaskEmbed.

**Augmentation compression.** MaskEmbed can be viewed as compressing a large number of augmentations into a single learned representation: we query specific augmentations based on how the embeddings are masked, and we obtain approximate reconstructions via the decoder. We note that CLIPSelf (Wu et al., 2024) can also be viewed as a form of augmentation compression with crops rather than masks.

**Relationship to masked image modeling.** MaskEmbed bears some similarity to BERT-style masked imaging modeling (MIM) methods like MAE, MaskFeat and BEiT (He et al., 2022; Wei et al., 2022; Bao et al., 2021), but there are several important differences. 1) When encoding images, MIM methods mask the image at the input layer; MaskEmbed encodes the entire image and masks only at the output embedding layer. 2) MIM methods adopt static labels for each patch (although they typically only train on masked patches); we do not require labels for each patch embedding, and instead supervise predictions via their ability to reconstruct arbitrary masked teacher views. 3) Most MIM methods are designed for pre-training; MaskEmbed is a post-training method that can be applied to any pre-trained ViT backbone, including strong pre-training approaches that MIM methods struggle to match (e.g., CLIP, SigLIP; Radford et al. 2021; Zhai et al. 2023b).

**Relationship to feature attribution.** As described in the main text, our reconstruction objective in Equation (1) generalizes an existing feature attribution approach (Jethani et al., 2021; Covert et al., 2022). Given masked outputs $f(m(x)) \in \mathbb{R}^d$ and a learned patch embedding model $g_\theta(x) \in \mathbb{R}^{n \times d}$, we can train the model to approximate $m^\top g_\theta(x) \approx f(m(x))$ for all $m$ using the following objective:

$$\min_\theta \ \mathbb{E}_{x,m}\left[\left\|m^\top g_\theta(x) - f\big(m(x)\big)\right\|^2\right]. \tag{2}$$

Unlike in our generalization that uses an expressive decoder, the resulting patch embeddings from Equation (2) have an analytic solution: the solution depends on the choice of mask distribution $p(m)$, and there exists a specific distribution that results in Shapley values (Charnes et al., 1988). Additionally, the learned embeddings share the semantics of the original model: for example, if $f(x)$ is a classifier, then the learned embeddings represent how each patch affects the class probabilities. Our generalization sacrifices these properties, but we find that this is necessary to learn richer patch embeddings.

**Relationship to hybrid ViTs and convolutional patch embeddings.** The original ViT architecture uses a lightweight linear projection to turn patches into tokens, and then passes these through a series of transformer blocks (Dosovitskiy et al., 2020). Other works have explored using more expressive patch embedding modules, e.g., a series of residually connected convolutions (Xiao et al., 2021). The combined model $h_\phi(g_\theta(x))$ we train with MaskEmbed can be viewed as using a highly expressive, transformer-based patch embedding module followed by a small number of transformer blocks that aggregate the rich patch embeddings. If this architecture were trained directly on a prediction task like image classification, the intermediate embeddings would not be constrained to be patch-specific; they are only forced to represent localized semantics in our approach because 1) we mask at the internal embedding layer, and 2) we use labels that change depending on the mask.

**Objective degeneracy.** One potential concern about our approach is that the objective in Equation (1) is degenerate: it contains a trivial solution where the encoder acts as an identity function and the decoder replicates the teacher model, or $g_\theta(\cdot) = I(\cdot)$ and $h_\phi(\cdot) = f(\cdot)$. This solution is undesirable because it fails to encode rich semantics in each patch embedding, and when training a VLM it is equivalent to passing raw patch projections (similar to the Fuyu architecture; Bavishi et al. 2023). Given the strong performance we observe in practice from MaskEmbed, we reason that the trivial solution is avoided due to 1) the encoder's strong initialization, and 2) the decoder's small number of parameters and weak initialization. We tried training the encoder from scratch in our early experiments, and we found that it was important to use a shallow decoder to avoid simply preserving information with the encoder and offloading computation. However, the objective degeneracy does not appear to be an issue when fine-tuning.

**Need for self-attention.** A related observation is that because we only need patch-specific information in each learned embedding to reconstruct masked views, we may not need self-attention in the encoder. For example, a helpful inductive bias could be to replace the ViT transformer blocks with residually connected MLPs, because this prevents patches from mixing information. We experimented with such an architecture and found that it performed poorly, learning more slowly and converging to a worse model than a ViT encoder even when both were trained from scratch. Interestingly, this suggests that inter-patch communication is helpful to understand each patch's semantics, and it shows that the expressive ViT architecture is highly beneficial for this task.

# E   VLM EXPERIMENT DETAILS & ADDITIONAL RESULTS

**Training recipe.** Following Karamcheti et al. (2024), we train the VLM in a single stage with the ViT frozen. This differs from some works that fine-tune the vision backbone and/or include a preliminary training stage to only train the vision-language adapter, including Qwen-VL (Bai et al., 2023), Idefics2 (Laurençon et al., 2024), DeepSeek-VL (Lu et al., 2024) and Pali-Gemma (Beyer et al., 2024). We use these settings because they were found to work best in this training library and with our quantity of training data.

**Hyperparameters.** Our hyperparameters are identical to those in Karamcheti et al. (2024), which themselves are inspired by Llava-1.5 (Liu et al., 2024). We report these below in Table 9. All VLMs are trained on a single node with 8 NVIDIA A100 SXM4 80GB GPUs.

Table 9: **VLM training hyperparameters.**

| Hyperparameter | Value |
| --- | --- |
| Epochs | 2 |
| Global batch size | 128 |
| Max sequence length | 2048 |
| Weight decay | 0.1 |
| Gradient clipping | 1.0 |
| Optimizer | AdamW |
| $\beta_1, \beta_2$ | (0.9, 0.999) |
| Learning rate schedule | Linear warmup + cosine decay |
| Max learning rate | 2e-5 |
| Min learning rate | 0 |
| Warmup ratio | 0.03 |

**Training data mixture.** The Llava-1.5 training data mixture (Liu et al., 2024) consists of data sourced from several pre-existing datasets. These include synthetic instruction completions from the original Llava work (Liu et al., 2023c), a collection of existing VQA datasets (VQAv2, GQA, OCR-VQA, OK-VQA, A-OKVQA; Goyal et al. 2017; Hudson & Manning 2019; Marino et al. 2019; Mishra et al. 2019; Schwenk et al. 2022), captioning data (TextCaps; Sidorov et al. 2020), referring expression data (RefCOCO, Visual Genome; Kazemzadeh et al. 2014; Yu et al. 2016; Krishna et al. 2017), and ShareGPT data sourced from user conversations (ShareGPT, 2023). Our extended data mixture also includes the recent LVIS-Instruct-4V (Wang et al., 2023a) and LRV-Instruct (Liu et al., 2023b) datasets, which roughly double the number of training examples.

**Benchmarks.** Our benchmarks are summarized in Table 10, including the prompt type, scoring method and details about variants of certain tasks. Some benchmarks are scored based on exact match using model response probabilities, others use intersection-over-union (IoU) thresholds for bounding box predictions, and others use the standard VQA scoring method (Antol et al., 2015). All our reported results use full splits set up by Karamcheti et al. (2024) consisting of several thousand examples each. Our radar charts use axes that are scaled separately for each benchmark based on the mean and standard deviation of performance within our pool of models; the models in this pool include the main runs with the original and locality-aligned backbones (Figure 5), ablations on the vision-language adapter described below (Figure 11), and DINOv2 feature fusion (Figure 13), all for both the CLIP and SigLIP backbones.

Table 10: **Summary of VLM benchmarks.**

| Benchmark | # Examples | Prompt Type | Scoring | Details |
|---|---|---|---|---|
| VizWiz | 4319 | Open-ended | VQA | Some questions are unanswerable |
| VQAv2 | 214354 | Open-ended | VQA | |
| GQA | 12578 | Open-ended | Exact match | |
| TextVQA (ocr) | 5000 | Open-ended | VQA | Prompt includes OCR dump |
| TextVQA (pure) | 5000 | Open-ended | VQA | No OCR dump |
| AI2D | 15501 | Multiple choice (4) | Exact match | |
| RefCOCO | 10834 | Bounding box | Acc @ 0.5 IoU | Spatial terms allowed |
| RefCOCO+ | 10758 | Bounding box | Acc @ 0.5 IoU | No spatial terms allowed |
| RefCOCOg | 4896 | Bounding box | Acc @ 0.5 IoU | Long object descriptions |
| OCID-Ref | 18342 | Bounding box | Acc @ 0.25 IoU | |
| VSR | 1222 | True/false | Exact match | |
| TallyQA (complex) | 15598 | Multiple choice (16) | Exact match | Involve filtering criteria |
| TallyQA (simple) | 22991 | Multiple choice (16) | Exact match | No filtering criteria |
| POPE | 9000 | Open-ended | Exact match | |

### E.1 ADDITIONAL RESULTS

We now report several additional results from our VLM experiments.

First, Figure 11 shows a series of ablations for VLMs trained using different vision-language adapters. In the main text, we report that using the standard MLP adapter for aligned backbones degrades performance (see "Aligned MLP" vs. "Baseline MLP") but that using the decoder as an adapter improves performance (see "Aligned Decoder"). To be sure that our improvements are due to locality alignment and not only the stronger adapter, we run several experiments using different adapter approaches for the baseline ViTs. First, we try training a transformer adapter from random initialization with the same size as the aligned model's decoder; we find that this hurts performance compared to the MLP adapter (see "Baseline Transformer"), and we suspect that this is due to our VLM setup having insufficient training data to learn this module from random initialization. Previous works that successfully use transformer-based adapters have significantly more training data (Bai et al., 2023; Laurençon et al., 2024), so this result suggests that the decoder adapter is effective in part because it is initialized from pre-trained parameters.

Next, because a fair comparison with our aligned model's decoder is not possible for the baseline backbone, we attempt to mimic the idea of using pre-trained transformer layers for the adapter: we simply use the last two ViT blocks with an additional linear layer, which we refer to as a *truncated* adapter. We remark that this represents partially fine-tuning the backbone, which along with training it using low-rank updates (Laurençon et al., 2024), unfreezing it partway through training (Lu et al., 2024), and giving it a longer warmup schedule (Beyer et al., 2024) is an option to stabilize joint fine-tuning. We find that this approach is less effective than the decoder adapter for aligned models (see "Aligned Truncated" vs. "Aligned Decoder"), but that it can improve performance over a MLP adapter for the baseline model (see "Baseline Truncated" vs. "Baseline MLP").

Since this is a new stronger baseline, we show a head-to-head comparison with our locality-aligned approach in radar charts in Figure 12. We find that the locality-aligned models preserve their improvements in several tasks, including AI2D and all three RefCOCO variants for both models, as well as POPE and TallyQA (Simple) for CLIP ViT-L @ 336px and VizWiz and OCID-Ref for SigLIP SO400M @ 384px. Overall, we conclude that our adapter strategy explains some of the gains observed in Figure 5, but that even adjusting for this with a stronger baseline shows improvements in several tasks, especially object localization and chart understanding.

Finally, Figure 13 and Figure 14 show results from our feature fusion runs with DINOv2 (Oquab et al., 2023; Darcet et al., 2023). Our implementation of feature fusion follows Karamcheti et al. (2024): we concatenate the two output sequences along their embedding dimension and then pass this through a MLP adapter. As we describe in the main text, the fused backbones often lead to larger gains in core localization tasks, likely due to DINOv2's excellent performance at dense prediction tasks (Oquab et al., 2023); however, it also leads the model to degrade in other ways, notably in VizWiz and TextVQA, which does not occur with our locality-aligned backbones. Overall, the more robust improvements from locality alignment make it an appealing option to improve localization tasks without negatively impacting other model capabilities.

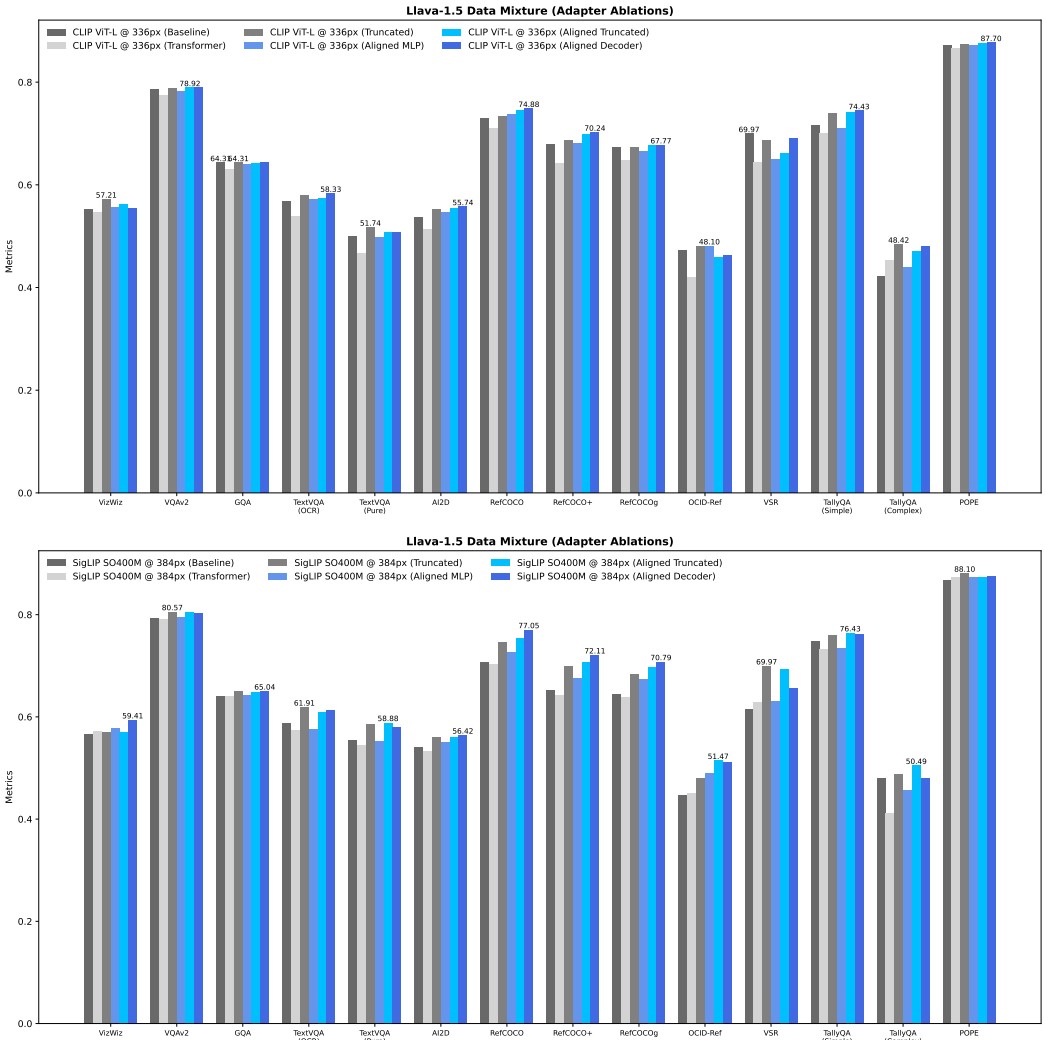

Figure 11: **VLM adapter ablations.** We report results for several vision-language adapter ablations using both the baseline and locality-aligned backbones.

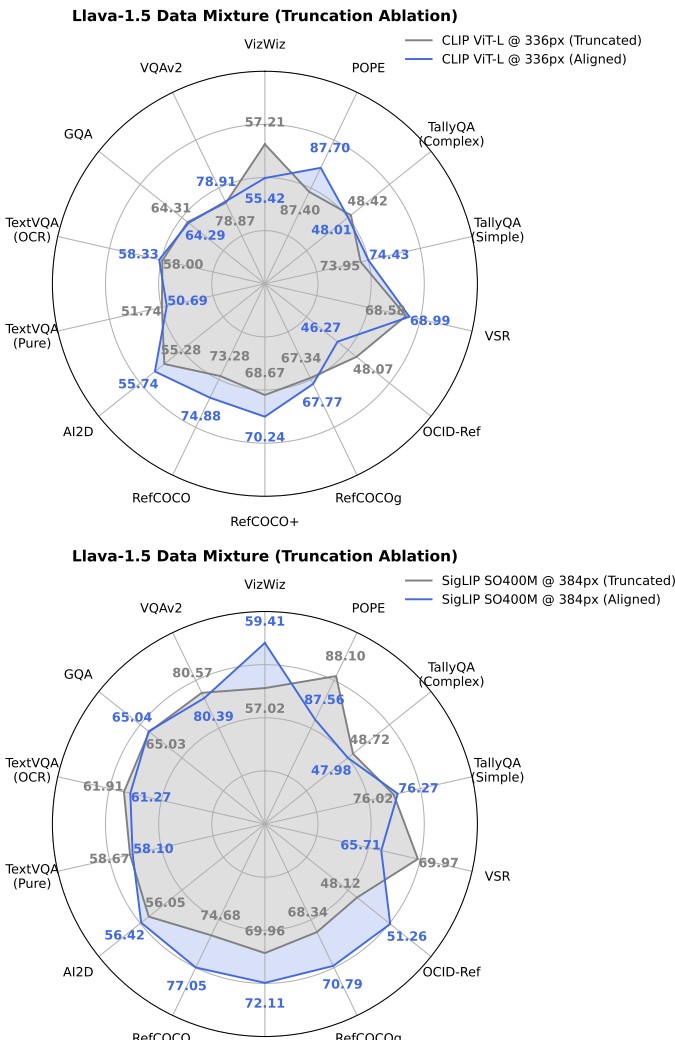

Figure 12: **Comparison between locality alignment and original model with truncated adapter.** We find that VLMs trained with locality-aligned backbones often outperform a new and stronger baseline, which truncates the last two ViT layers and fine-tunes them as a vision-language adapter.

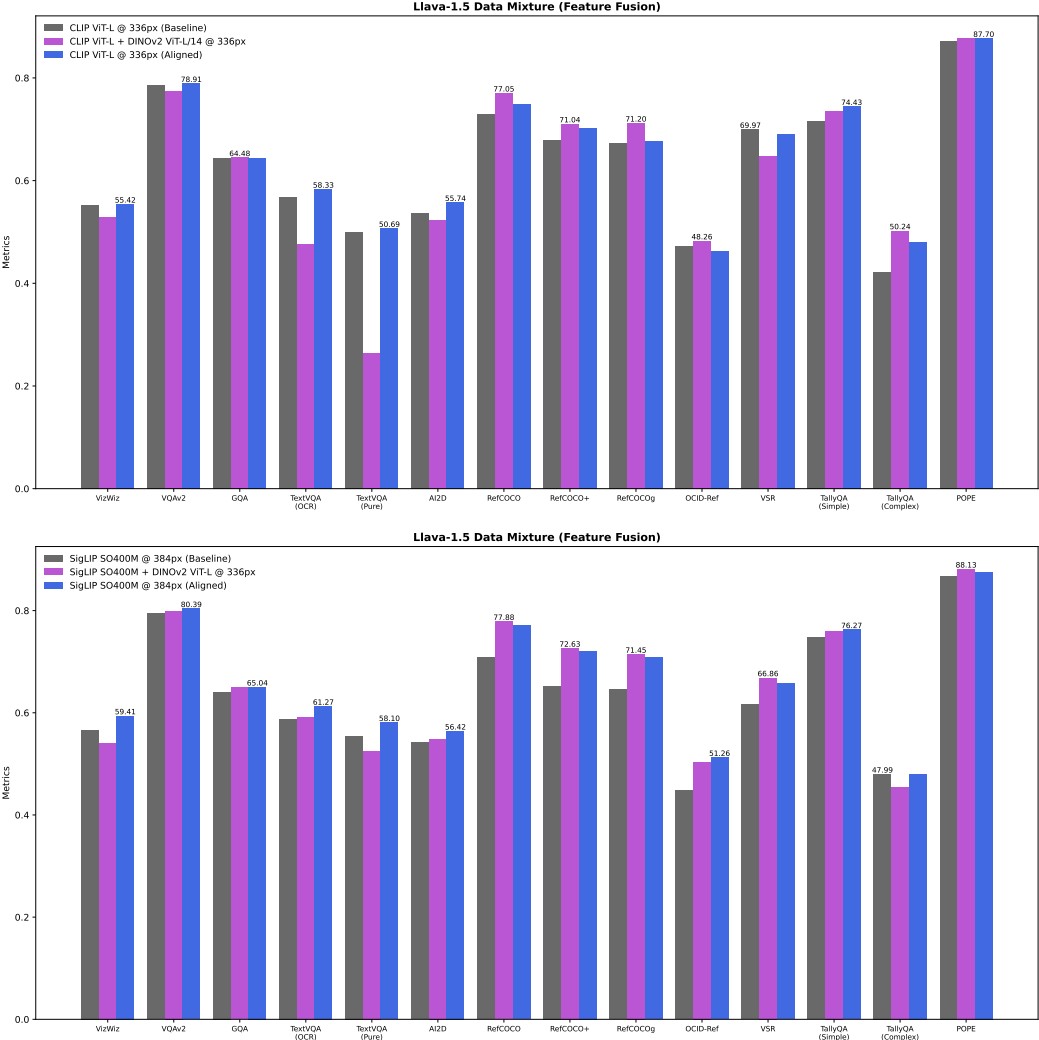

Figure 13: **VLM comparison with DINOv2 feature fusion.** We compare the baseline and locality-aligned VLMs with an alternative strategy to enhance the visual features, which is to fuse with DINOv2's output embedding. We find that this approach can lead to larger gains on localization tasks but also degrades the model in other ways.

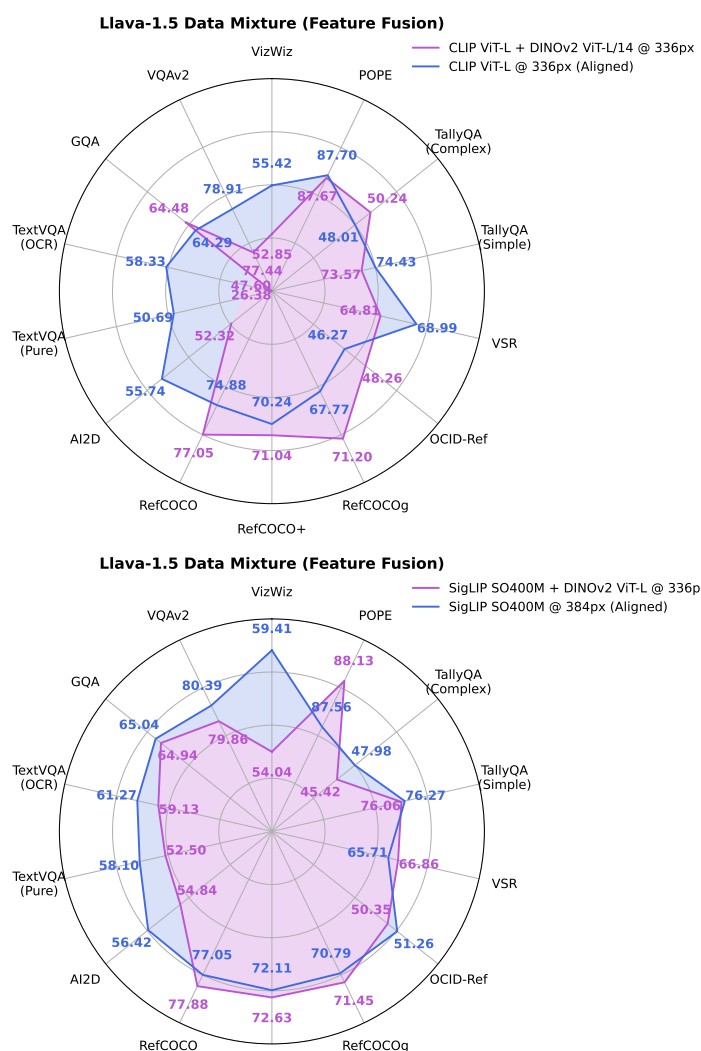

Figure 14: **VLM comparison with DINOv2 feature fusion.** We compare VLMs with locality-aligned backbones to fusing features between CLIP/SigLIP and DINOv2. TextVQA benchmarks are not shown for CLIP ViT-L + DINOv2 fusion due to the accuracy lying outside the display range, more than three standard deviations below the mean performance.

