# OpenReview forum: "Locality Alignment Improves Vision-Language Models"
_ICLR.cc/2025/Conference — ICLR 2025 Poster_

### Official Review · Reviewer_XvKA · 2024-10-26

**Soundness:** 4
**Presentation:** 4
**Contribution:** 4
**Rating:** 8
**Confidence:** 4

**Summary:**

This paper introduces a novel, computationally efficient mask-based self-supervised loss function designed to enhance the local feature representations of a pretrained Vision Transformer (ViT) initially trained on a global, image-level task. This approach aims to improve the ViT's utility for Vision-Language Model (VLM) training. By applying their method across various backbones, the authors demonstrate its generalizability and report gains in patch-level semantic segmentation and across multiple vision-language tasks.

**Strengths:**

Simplicity and novelty of the approach, with a clear explanation of the intuition behind the formulation of patch embeddings (g_i) and the task.

Demonstration of general applicability by using a variety of backbones trained on different tasks.

Comprehensive evaluation across diverse VLM benchmarks.

**Weaknesses:**

Ablation studies could be more comprehensive. Some, like the effects of data augmentation and training data scale, feel unnecessary or self-evident. Exploring a broader range of datasets, such as CC3M (diverse) versus IN1k or SAM images (multi-object), could have offered more insightful findings on generalizability.

Limited ablation of the loss function. For example, testing reconstruction of only unmasked tokens rather than the entire embedding sequence could provide valuable insights into the role of different token types in the loss.

Comparison with alternative masking strategies is missing. While the rationale for masking in the current way is sound, comparing with an approach like dBOT—where the student rather than the teacher is masked—could have strengthened their case, as dBOT follows a nearly identical pipeline and has shown strong spatial feature learning.

dBOT: Exploring Target Representations for Masked Autoencoders

**Questions:**

In the current setup, the entire embedding sequence is reconstructed during training. Have you considered ablations that reconstruct only unmasked tokens?

While I understand your reasoning for masking the teacher, have you explored or considered a comparison with approaches where the student is masked, as in the dBOT framework?

(This one is due to my lack of knowledge about single stage VLMs), Can you give more context on the significance of the VLM benchmark improvements? Some of the relative improvements on Figure 5 are so small that they seem like noise. (but again, it just might be duo to my lack of knowledge)

---

> ### Author Response · Authors · 2024-11-25
> **Response to XvKA (Part 1)**
>
> Thank you for taking the time to read our paper and provide helpful feedback! We respond below to the points raised in your review.
>
> > Ablation studies could be more comprehensive [...] Exploring a broader range of datasets, such as CC3M (diverse) versus IN1k or SAM images (multi-object), could have offered more insightful findings on generalizability.
>
> Thanks for raising this point. Investigating the role of data scale/diversity is an important direction for future work, and we made sure to mention it in our conclusion. We didn’t prioritize data ablations here because our main goal was to show that locality alignment works, and we found that it does even with relatively small datasets (by modern standards) like IN1k and IN21k. It may turn out to work even better with other datasets, but investigating a range of options is difficult given our computational resources and the short rebuttal period, so we are deferring this study to future work. In addition to CC3M and the SAM dataset, we are also interested in whether locality alignment can benefit from training longer on larger, more diverse datasets like DataComp-1B and DFN-2B.
>
> > Limited ablation of the loss function. For example, testing reconstruction of only unmasked tokens rather than the entire embedding sequence could provide valuable insights into the role of different token types in the loss.
>
> Thanks for the suggestion, we’re happy to include additional loss function ablations. Reconstructing either just the masked tokens or just the unmasked tokens is a reasonable idea because methods like MAE/BEiT do this, but it seems inadvisable here because all of the token reconstruction targets are non-trivial so ignoring any of them will only make the task easier. We tested this and found that both versions degrade local probing performance compared to a loss that considers all tokens (using our benchmark from Section 4).
>
> Our results are shown in the table below, where all experiments are conducted with IN1k ViT-B models for consistency with our other ablations. We also included a comparison with cosine similarity loss and L1 loss, neither of which outperforms our choice of MSE. We will include these results in the revised paper.
>
> | Loss variant | Local | Global |
> | ---          | ---   | ---    |
> | Cosine       | 45.55 | 51.37  |
> | L1           | 45.26 | 51.10  |
> | MSE          | 45.66 | 51.43  |
> | Masked MSE   | 42.48 | 45.39  |
> | Unmasked MSE | 45.00 | 51.67  |
>
> > Comparison with alternative masking strategies is missing. While the rationale for masking in the current way is sound, comparing with an approach like dBOT—where the student rather than the teacher is masked—could have strengthened their case, as dBOT follows a nearly identical pipeline and has shown strong spatial feature learning.
>
> Thanks for the suggestion. Reviewer ARqQ also requested a comparison with dBOT, and we provided a longer discussion there of our findings and attempts to use the dBOT codebase. Briefly, the main points are:
>
> 1. dBOT is another masked image modeling (MIM) method like MAE, BEiT, MaskFeat and EVA02, and these have important differences with our proposal. The key differences between MaskEmbed and MIM methods as a whole are 1) we mask the student model at its *output layer* rather than its input layer, 2) we mask the teacher as well to create reconstruction targets that vary with the mask $m$. These differences encourage MaskEmbed to learn specific features associated with each patch, whereas MIM teaches a model to impute features for missing patches. We’ve clarified these differences in our revised Appendix C.1 under a section called “Relationship with masked image modeling,” and we also have a new extended related work section in Appendix A.
>
> 2. We tested dBOT’s random teacher checkpoint in our probing benchmark, and found that it’s slightly better than MAE, but not competitive with language-supervised models. This is dBOT’s main proposal, and it leaves a lot of room for improvement on this evaluation.
>
> 3. We tried to use dBOT’s CLIP teacher checkpoint, but we were unable to get reasonable results despite our best attempts to follow the authors’ code. Fortunately, because dBOT’s teacher bootstrapping technique didn’t work for CLIP (as described in their Appendix C), their checkpoint uses standard single-stage MIM, and a reasonable surrogate is to compare to other CLIP MIM models. We consider EVA02 ViT-B from Fang et al. (2023), which was trained using MIM with a large EVA-CLIP 1B teacher, and should therefore have an advantage over our locality-aligned CLIP ViT-B. While EVA02 ViT-B outperforms the pre-trained CLIP ViT-B, our locality-aligned version has stronger performance at both local and global probing. Interestingly, locality alignment also leads to a significant improvement on EVA02’s local probing performance, showing that MIM does not saturate its performance at this task.

---

> ### Author Response · Authors · 2024-11-25
> **Response to XvKA (Part 2)**
>
> The following table shows the relevant results from the probing benchmark:
>
> | Model                | Teacher     | Local | Global |
> | ---                  | ---         | ---   | ---    |
> | MAE ViT-B            | -           | 39.46 | 43.53  |
> | dBOT ViT-B           | Random      | 41.57 | 44.12  |
> | CLIP ViT-B           | -           | 44.63 | 52.61  |
> | CLIP ViT-B (Aligned) | CLIP ViT-B  | 46.32 | 54.55  |
> | dBOT ViT-B           | CLIP ViT-B  | 4.67  | 7.00   |
> | EVA02 ViT-B          | EVA-CLIP 1B | 44.91 | 52.93  |
> | EVA02 ViT-B (Aligned)| EVA02 ViT-B | 49.21 | 51.47  |
>
>
> > Can you give more context on the significance of the VLM benchmark improvements? Some of the relative improvements on Figure 5 are so small that they seem like noise.
>
> Thanks for raising this point. We set the axis scales for our radar charts following the approach from Karamcheti et al. (2024): we considered the pool of all VLMs we evaluated and set axes using the mean +/- 3 times the standard deviation for each benchmark. This leads to reasonable scaling for most benchmarks, but we agree that there are a couple whose differences seem magnified due to low performance variability (POPE and GQA). We thought it best to keep our scaling method consistent between benchmarks rather than set these differently by hand.
>
> Based on your comment, we’ve made sure this point is explained in the paper. Section 5.1 now contains the following sentence: “Following prior work (Karamcheti et al., 2024), we scale each benchmark’s y-axis based on the mean and standard deviation within our pool of models.” We also explain this point in more detail in Appendix D, including the specific pool of models used for axis scaling.

---

> > ### Comment · Reviewer_XvKA · 2024-11-26
> >
> > Thanks for your answers, I have raised my score from 6 to 8.

---

> > > ### Author Response · Authors · 2024-11-26
> > > **Thank you**
> > >
> > > Thanks again for your feedback and the prompt reply, we appreciate the score raise! Best of luck.

---

### Official Review · Reviewer_ARqQ · 2024-11-02

**Soundness:** 3
**Presentation:** 3
**Contribution:** 2
**Rating:** 5
**Confidence:** 4

**Summary:**

This paper proposes a  MaskEmbed fine-tuning procedure that uses a masked reconstruction loss to improve the path-level semantics of the visual encoder. Experiment results show that the proposed method can improve performance across a range of VLM benchmarks.

**Strengths:**

1. The paper is well-written and easy to follow.
2. The proposed MaskEmbed training diagram is effective in learning local semantics.

**Weaknesses:**

1. The authors need to include comparisons and discussions with more methods, such as dBOT[1] and UMG-CLIP[2].

a) dBOT[1] employs a distillation strategy similar to the method presented in the paper, so it is necessary to discuss the differences with this method and provide performance comparisons.

b) UMG-CLIP[2] directly incorporates fine-grained annotations to enhance CLIP's Locality Alignment. I am curious whether the proposed method has advantages over this method in some fundamental visual perception benchmarks.  Besides, is it possible to use a similar visualization approach as in UMG-CLIP to further illustrate the locality of the features in MaskEmbed?

2. The experimental comparisons are not sufficient.

a) As previously mentioned, it is necessary to supplement more results on visual perception benchmarks.

b) The paper employs a one-stage strategy for training VLMs. As far as I know, the majority of current VLM methods employ a multi-stage training approach, and the mentioned Llava-1.5 does as well, which gives it better performance than the one-stage baseline in this paper. I hope the authors can train the VLM model according to the Llava-1.5 setup and conduct performance comparisons on more benchmarks used in Llava-1.5 (r.f. Benchmarks in Table 3 and Table 4 of Llava-1.5).

c) To my knowledge, the latest method in the Llava series, Llava-OneVision[3], fine-tunes the vision encoder part simultaneously during VLM training. Many other recent methods[4][5] also adopt this setting, making the claim in lines 52-54 somewhat Inadequate. I wonder whether simultaneously fine-tuning the vision encoder could lead to VLMs gaining local semantic understanding, potentially diminishing the advantages of the proposed method.

[1] Exploring target representations for masked autoencoders.

[2] UMG-CLIP: A Unified Multi-Granularity Vision Generalist for Open-World Understanding.

[3] LLaVA-OneVision: Easy Visual Task Transfer.

[4] Qwen2-VL: Enhancing Vision-Language Model’s Perception of the World at Any Resolution.

[5] MiniCPM-V: A GPT-4V Level MLLM on Your Phone.

**Questions:**

Please refer to the 'Weaknesses' part.

---

> ### Author Response · Authors · 2024-11-25
> **Response to ARqQ (Part 1)**
>
> Thank you for taking the time to read our paper and provide helpful feedback! We respond below to the points raised in your review.
>
> > The authors need to include comparisons and discussions with more methods, such as dBOT[1] and UMG-CLIP[2]. dBOT[1] employs a distillation strategy similar to the method presented in the paper, so it is necessary to discuss the differences with this method and provide performance comparisons.
>
> Thanks for pointing us to dBOT, we’ve carefully reviewed the paper and will be sure to cite it in our work. dBOT is another masked image modeling (MIM) method like MAE, BEiT, MaskFeat and EVA02, and has important differences with our proposal. The key differences between MaskEmbed and MIM methods as a whole are 1) we mask the student model at its *output layer* rather than its input layer, 2) we mask the teacher as well to create reconstruction targets that vary with the mask $m$. These differences encourage MaskEmbed to learn specific features associated with each patch, whereas MIM teaches a model to impute features for missing patches. Based on your comment, we’ve clarified these differences in our revised Appendix C.1 under a section called “Relationship with masked image modeling,” and we also have a new extended related work section in Appendix A.
>
> We’re also happy to include results for dBOT. We’ll separately describe our attempts to use dBOT’s checkpoints trained with the random teacher and CLIP teacher below:
>
> - dBOT’s random teacher checkpoint was straightforward to load from the official [GitHub repository](https://github.com/liuxingbin/dbot?tab=readme-ov-file), and we evaluated it using our probing benchmark from Section 4. The results are slightly better than MAE, but this backbone is not competitive with language-supervised models like CLIP. This shows that dBOT’s main proposal leaves large room for improvement on this benchmark. With locality alignment, we can start from a strong model like CLIP and make it even stronger with a short fine-tuning phase.
>
> - dBOT’s CLIP teacher checkpoint would have been more interesting, because MIM and locality alignment are competing interventions to improve a strong pre-trained model like CLIP. Unfortunately, we were unable to get reasonable results with the model provided by the authors (described in more detail below). However, because dBOT’s proposal of bootstrapping the teacher didn’t work for CLIP (as shown in their Appendix C), their CLIP-teacher checkpoint used standard single-stage MIM, and a reasonable surrogate is to compare with other MIM methods that use CLIP teachers. One such model is EVA02 from Fang et al. (2023), which has the advantage of using a stronger EVA-CLIP 1B teacher. EVA02 was already included in our experiments (Table 5), and we find that it’s indeed better than the pre-trained CLIP ViT-B, but locality alignment leads CLIP to exceed EVA02 at both local and global probing. Interestingly, we also find that locality alignment also significantly improves EVA02’s local probing performance, suggesting that MIM alone does not saturate its performance at this task.
>
> The following table shows the relevant results from the probing benchmark:
>
> | Model                | Teacher     | Local | Global |
> | ---                  | ---         | ---   | ---    |
> | MAE ViT-B            | -           | 39.46 | 43.53  |
> | dBOT ViT-B           | Random      | 41.57 | 44.12  |
> | CLIP ViT-B           | -           | 44.63 | 52.61  |
> | CLIP ViT-B (Aligned) | CLIP ViT-B  | 46.32 | 54.55  |
> | dBOT ViT-B           | CLIP ViT-B  | 4.67  | 7.00   |
> | EVA02 ViT-B          | EVA-CLIP 1B | 44.91 | 52.93  |
> | EVA02 ViT-B (Aligned)| EVA02 ViT-B | 49.21 | 51.47  |
>
> Re: loading dBOT CLIP-teacher checkpoint. We loaded the checkpoint using the authors’ [beit branch](https://github.com/liuxingbin/dbot/tree/beit), which provides separate modeling code for pre-training and fine-tuning, with checkpoint loading code in their `modeling_finetune.py` script. The probing benchmark results were worse than the dBOT random teacher checkpoint, so we suspected an issue and took several steps to identify it. We ensured that 1) the model was initialized with the correct arguments, 2) all relevant keys were loaded into the model, 3) there were no clear inconsistencies in the pre-training and fine-tuning modeling code, 4) the `forward_features` outputs were identical when the checkpoint was loaded into the pre-training and fine-tuning models. We couldn’t identify the problem with the checkpoint, but we believe our EVA02 comparison provides reasonable evidence that MIM with a pre-trained teacher is not a substitute for locality alignment.

---

> > ### Author Response · Authors · 2024-11-25
> > **Response to ARqQ (Part 2)**
> >
> > > UMG-CLIP[2] directly incorporates fine-grained annotations to enhance CLIP's Locality Alignment. I am curious whether the proposed method has advantages over this method in some fundamental visual perception benchmarks.
> >
> > Thanks for pointing us to the UMG-CLIP paper, we’ll be sure to cite it as well. The main difference is that UMG-CLIP synthesizes dense supervision from a range of existing models (SAM, ViTDet, BLIP2, MaskDINO, Shikra, RAM, EVA, etc) and uses this dataset to train a new CLIP-like model from scratch, whereas we can fine-tune any existing model with no need for external supervision. The advantages of our method are its simplicity (no complex data curation and filtering), efficiency (alignment can be short compared to pre-training), self-containedness (no need for supervision from ~10 external models) and scalability (we can train on an arbitrary amount of unlabeled data). UMG-CLIP’s main advantage, which frankly seems quite useful, is that it learns from multiple strong teachers. An improved future version of MaskEmbed could incorporate this idea from UMG-CLIP, perhaps by learning multiple decoders to predict masked views from multiple teachers.
> >
> > > Besides, is it possible to use a similar visualization approach as in UMG-CLIP to further illustrate the locality of the features in MaskEmbed?
> >
> > Thanks for the suggestion. It seems like you’re referring to Figure 5 in the UMG-CLIP paper, where the authors visualize the similarity between patch embeddings and text embeddings. This visualization approach doesn’t work for locality-aligned models because our patch embeddings are trained to reconstruct masked teacher views, not have high similarity with matching captions. We reason that this is acceptable because our eventual use case is using the backbone for downstream tasks like training VLMs, and this doesn’t require interpretable visualizations.
> >
> > Interestingly, such visualizations are possible under the additive version of our approach outlined in Section 3.1. We discuss this point in our updated Appendix C.1, where we explain the relationship with feature attribution and the known closed-form solutions for patch embeddings in that setup. However, we don’t use this approach because it’s less effective than our Section 3.2 formulation for enhancing models’ local feature extraction ability (see Table 1).
> >
> > > As previously mentioned, it is necessary to supplement more results on visual perception benchmarks.
> >
> > Thanks for this suggestion. Based on similar comments from other reviewers, we’ve added new results fine-tuning models for IN1k before and after locality alignment to test its impact on global semantic understanding. Our results show that performance doesn’t degrade but instead improves after locality alignment, likely because our objective encourages the model to retain the original prediction $f(x)$ whenever we sample the all-ones mask $m = 1$.
> >
> > Our IN1k fine-tuning results for CLIP models of different scales are shown in the following table, and we’ll include them in our revised submission. Our hyperparameters are chosen to match those in OpenCLIP. In addition to achieving better final results, locality-aligned models also trained significantly faster in the early epochs, suggesting that they learn strong features that are easy to adapt with fine-tuning.
> >
> > | Model              | Baseline | Aligned |
> > | ---                | ---      | ---     |
> > | CLIP ViT-B         | 82.4     | 82.9    |
> > | CLIP ViT-L         | 84.4     | 85.4    |
> > | CLIP ViT-L @ 336px | 85.0     | 85.5    |

---

> ### Author Response · Authors · 2024-11-25
> **Response to ARqQ (Part 3)**
>
> > As far as I know, the majority of current VLM methods employ a multi-stage training approach, and the mentioned Llava-1.5 does as well, which gives it better performance than the one-stage baseline in this paper.
>
> Thanks for raising this point. Our training runs used the approach recommended in the Prismatic VLMs codebase (Karamcheti et al., 2024), which is essentially a Llava-1.5 replication that carefully studied key hyperparameters. One of their experiments tested the value of an initial adapter training stage, and they found that it slightly degraded the final model compared to a one-stage approach (see their Figure 4). We found this surprising but confirmed it in our own early experiments. Based on this, we proceeded with the simpler one-stage training approach. We’re aware that this is somewhat uncommon in the literature, but it didn’t seem justifiable to use a worse-performing setup just to be consistent with Llava/Llava-1.5. In any case, this choice doesn’t seem like it would interact strongly with the choice of vision backbone.
>
> > To my knowledge, the latest method in the Llava series, Llava-OneVision[3], fine-tunes the vision encoder part simultaneously during VLM training. Many other recent methods[4][5] also adopt this setting, making the claim in lines 52-54 somewhat Inadequate. I wonder whether simultaneously fine-tuning the vision encoder could lead to VLMs gaining local semantic understanding, potentially diminishing the advantages of the proposed method.
>
> Thanks for raising this point. It’s true that some recent VLMs fine-tune the vision backbone during training, including DeepSeek-VL, Qwen2-VL, PaliGemma and Llava-OneVision. We didn’t do so because it appears to be harmful when training with our scale of multi-modal data (~1M examples), and our choice is consistent with the Llava/Llava-1.5/Prismatic setups that all train at our data scale. However, it’s true that the effect of our backbone could be different when performing full fine-tuning. We reason that it would still help to start from a better initialization (results like this are shown for SigLIP vs DINOv2 in Cambrian-1), but we’re unable to perform the large-scale experiments to verify this. Based on your comment, we’ll be sure to acknowledge this limitation in the paper. Our conclusion contains the following text: "The benefits of locality alignment may change with end-to-end fine-tuning, but we did not explore this because it is unhelpful with our quantity of multi-modal training data (Karamcheti et al., 2024). An important direction for future work is to test locality alignment in other VLM training approaches, with larger LMs, and to evaluate how it composes with other techniques that enhance visual features."

---

> > ### Author Response · Authors · 2024-12-02
> > **Discussion reminder**
> >
> > Thanks again for your helpful feedback. As we approach the end of the discussion phase, we hope you can read our rebuttal and let us know if your concerns have been resolved. We believe we've addressed your request for comparison with prior works (UMG-CLIP and MIM methods like dBOT), results on visual perception benchmarks (IN1k), and questions about how we deviate from recent VLM training protocols. We are happy to discuss any of these points further if it would be helpful.

---

### Official Review · Reviewer_oNgu · 2024-11-04

**Soundness:** 2
**Presentation:** 3
**Contribution:** 2
**Rating:** 6
**Confidence:** 4

**Summary:**

This paper proposes improving the performance of Vision-Language Models (VLMs) on region-level visual tasks by enhancing their **locality alignment**. The authors argue that the poor performance of many current VLMs on spatial reasoning tasks can be attributed to the weak locality alignment of vision models, which is caused by image-level supervision and minimal inductive biases. To address this, the paper introduces a post-training approach called **MaskEmbed**, which aims to improve the locality of features. Specifically, MaskEmbed applies the same mask to both the input image of the teacher model and the output features of the encoder model. The masked features are then aligned with the teacher model’s output features through a decoder, thereby enhancing the encoder’s locality alignment. Both visualizations and experimental results demonstrate that MaskEmbed effectively improves locality alignment and boosts performance on downstream tasks.

**Strengths:**

The proposed method is relatively simple and provides a notable performance improvement.

**Weaknesses:**

1. **Issues with the Main Claim**:
   The paper’s primary claim is confusing. It begins by hypothesizing that VLMs perform poorly on region-level tasks due to image-level supervision and minimal inductive biases. However, pre-training methods like DINO, despite using image-level supervision, exhibit strong locality, to the point where their features can even be directly used for semantic segmentation maps. This suggests that the initial assumption may be flawed. Additionally, regarding the claim about minimal inductive biases, is the paper referring to the lack of inductive biases in ViT architectures? If so, would using a convolution-based structure like ConvNext or a hierarchical structure like Swin Transformer resolve this issue? The paper fails to sufficiently justify this claim, making the motivation behind **MaskEmbed** unclear. Consequently, **MaskEmbed** may not be effective in the aforementioned scenarios.

2. **Methodological Concerns**:
   There are also some issues with the methodology. The goal of **MaskEmbed** is to make the tokens output by the encoder—when the entire image is input—align with the tokens output by the teacher model when only partially visible patches are input. This approach could weaken the encoder's ability to model global interactions, potentially limiting its capacity to capture long-range dependencies. Such a strategy could harm performance on tasks requiring global semantic understanding. Furthermore, this method may also negatively impact region-level tasks if the masking approach crosses a certain threshold, as it constrains the model’s capacity for representation learning. These factors suggest that **MaskEmbed** may be highly sensitive to hyperparameters, which significantly limits its overall contribution.

**Response to the rebuttal**

Thank you to the authors for their comprehensive and detailed responses—most of my concerns have been resolved.

However, regarding the use of models with stronger locality, such as DINO and MAE, as vision encoders, the authors avoided addressing this issue by pointing out that most VLMs use CLIP/SigCLIP as their vision encoder. Nevertheless, many recent MLLM studies often incorporate mixed vision encoders, as demonstrated in [1,2]. Therefore, it would still be valuable to verify the effectiveness of the proposed method when applied to models with stronger locality.

In addition, although this work focuses on the VLM domain, its techniques primarily target vision encoders. To better highlight its advantages, I suggest integrating the proposed method into some vision pre-trained models, such as MAE or DINO, and evaluating whether the optimized models exhibit advantages in dense prediction tasks. This would better showcase the contributions of this work.

Despite some minor concerns, I find this to be an interesting study overall, and I have raised my score accordingly.

References:

[1] Jiang D, Liu Y, Liu S, et al. From clip to dino: Visual encoders shout in multi-modal large language models. arXiv preprint arXiv:2310.08825, 2023.

[2] Shi M, Liu F, Wang S, et al. Eagle: Exploring the design space for multimodal LLMs with mixture of encoders. arXiv preprint arXiv:2408.15998, 2024.

**Questions:**

1. How were the scales in **Figure 5** determined? The performance gains appear to be quite small, yet they are magnified in the figure, making it difficult to assess the actual level of improvement provided by this method.

---

> ### Author Response · Authors · 2024-11-25
> **Response to oNgu (Part 1)**
>
> Thank you for taking the time to read our paper and provide helpful feedback! We respond below to the points raised in your review.
>
> > However, pre-training methods like DINO, despite using image-level supervision, exhibit strong locality, to the point where their features can even be directly used for semantic segmentation maps. This suggests that the initial assumption may be flawed.
>
> Thanks for giving us the opportunity to clarify. A few models like DINO, DINOv2 and MAE exhibit relatively good locality, but this is because they’re trained with some form of dense supervision. Note that the dense supervision for MAE and DINOv2 is explicit, but for DINO it’s due to multi-crop training with random sub-images, see Caron et al. (2021) for more details. Unlike these models trained with dense supervision, other models trained with only image-level supervision have clear room for improvement in their ability to extract localized features, as we demonstrate by boosting their performance via locality alignment: our procedure uses only the model itself as supervision yet reliably improves local probing accuracy, and we show this for models including IN1k classifiers, CLIP, SigLIP, DFN, OpenCLIP and Moco v3 (see Figure 4 and Table 5).
>
> We believe our focus on these models trained with image-level supervision is warranted, because language-supervised models like CLIP/SigLIP are widely adopted in VLMs, whereas MAE and DINOv2 are not as effective (see Karamcheti et al., 2024; Tong et al., 2024). For these models that don’t excel at extracting localized semantics, we successfully address this shortcoming with locality alignment.
>
> > Additionally, regarding the claim about minimal inductive biases, is the paper referring to the lack of inductive biases in ViT architectures? If so, would using a convolution-based structure like ConvNext or a hierarchical structure like Swin Transformer resolve this issue?
>
> Thanks for raising this point. Yes, we’re referring to the lack of inductive bias in ViTs. We believe our focus on ViTs is justified given that they’re used for all recent state-of-the-art models and adopted in nearly all VLMs. However, you’re correct that a lack of local semantics could potentially be resolved by architectural solutions, such as reintroducing inductive biases like those in ConvNext or Swin Transformer, or even stronger options like an architecture that processes patches independently. It’s a reasonable idea that we also considered, but we’re more interested in solutions based on the training objective, because that lets us retain maximal expressive power and improve the best current models.
>
> Based on your comment, we’ve made sure this point is reflected in the paper. Our related work section contains the following sentence in the context of ViTs lack of localized features: “Some have proposed hybrid ViTs that reintroduce inductive biases from CNNs (Liu et al., 2021, Wu et al., 2021, d’Ascoli et al., 2021), but we improve the original ViT’s local feature extraction without sacrificing expressive power.”
>
> > This approach could weaken the encoder's ability to model global interactions, potentially limiting its capacity to capture long-range dependencies. Such a strategy could harm performance on tasks requiring global semantic understanding.
>
> Thanks for this suggestion, it’s a good idea to include experiments for global semantic understanding. We’ve added new results where we fine-tune models for IN1k classification, and we verified that performance does not degrade but instead improves after locality alignment. This is likely because when learning patch-specific features, the model is also trained to retain its original prediction: the all-ones mask $m = 1$ corresponds to reconstructing the original teacher output $f(x)$, so preserving global understanding is built into our objective.
>
> Our IN1k fine-tuning results for CLIP models of different scales are shown in the following table, and we’ll include them in our revised submission. Our hyperparameters are chosen to match those in OpenCLIP. In addition to achieving better final results, locality-aligned models also trained significantly faster in the early epochs, suggesting that they learn strong features that are easy to adapt with fine-tuning.
>
> | Model              | Baseline | Aligned |
> | ---                | ---      | ---     |
> | CLIP ViT-B         | 82.4     | 82.9    |
> | CLIP ViT-L         | 84.4     | 85.4    |
> | CLIP ViT-L @ 336px | 85.0     | 85.5    |

---

> > ### Author Response · Authors · 2024-11-25
> > **Response to oNgu (Part 2)**
> >
> > > Furthermore, this method may also negatively impact region-level tasks if the masking approach crosses a certain threshold, as it constrains the model’s capacity for representation learning. These factors suggest that MaskEmbed may be highly sensitive to hyperparameters, which significantly limits its overall contribution.
> >
> > Thanks for raising this concern. We included a set of ablations that helped guide hyperparameter selection (Appendix B.1 in our updated submission), and these include experiments related to the mask sampling distribution. It’s true that some mask distributions don’t work as well, but we found that a simple technique that works is to sample masks with cardinality chosen uniformly at random (called “uniform” in Table 4c). We use this approach in all our main experiments, and our results show that MaskEmbed reliably improves models’ region-level understanding ability.
> >
> > > How were the scales in Figure 5 determined? The performance gains appear to be quite small, yet they are magnified in the figure, making it difficult to assess the actual level of improvement provided by this method.
> >
> > Thanks for raising this point. We set the axis scales for our radar charts following the approach from Karamcheti et al. (2024): we considered the pool of all VLMs we evaluated and set axes using the mean +/- 3 times the standard deviation for each benchmark. This leads to reasonable scaling for most benchmarks, but we agree that there are a couple whose differences seem magnified due to low performance variability (POPE and GQA). We thought it best to keep our scaling method consistent between benchmarks rather than set these differently by hand.
> >
> > Based on your comment, we’ve made sure this point is explained in the paper. Section 5.1 now contains the following sentence: “Following prior work (Karamcheti et al., 2024), we scale each benchmark’s y-axis based on the mean and standard deviation within our pool of models.” We also explain this point in more detail in Appendix D, including the specific pool of models used for axis scaling.

---

> > > ### Author Response · Authors · 2024-12-02
> > > **Discussion reminder**
> > >
> > > Thanks again for your helpful feedback. As we approach the end of the discussion phase, we hope you can read our rebuttal and let us know if your concerns have been resolved. We believe we've addressed your questions about our premise, as well as your requests for masking ablations and evaluations of global semantic understanding. We are happy to discuss any of these points further if it would be helpful.

---

### Official Review · Reviewer_5BRv · 2024-11-04

**Soundness:** 2
**Presentation:** 2
**Contribution:** 2
**Rating:** 5
**Confidence:** 3

**Summary:**

The paper presents a method called Locality Alignment, which aims to improve the spatial reasoning capabilities of Vision Language Models (VLMs) by enhancing the local semantic understanding of pre-trained Vision Transformers (ViTs). Specifically,  the authors propose MaskEmbed, which uses a masked reconstruction loss to learn the semantic contributions of each image patch.

**Strengths:**

1. The paper is overall well-written.
2.  Locality alignment is efficient, requiring minimal additional computation compared to pre-training, making it a cost-effective solution.
3. The authors provide theoretical analysis and practical experiments to support their claims.

**Weaknesses:**

1. I suggest the authors to conduct a thorough analysis of MaskEmbed's sensitivity to hyperparameters. This includes varying mask sizes, patch sampling strategies, and the influence of different reconstruction targets. By understanding these sensitivities, the paper can provide guidelines for applying MaskEmbed effectively across various scenarios. Besides,  including additional evaluations that specifically test the impact of MaskEmbed on global semantic understanding tasks may help to validate whether the method indeed compromises the model's ability to capture long-range dependencies.
2. The methodology of MaskEmbed involves fine-tuning the encoder to align its output tokens with those of a teacher model when only partial patches of an image are visible. This approach, while innovative, raises concerns about its impact on the encoder's ability to model global interactions and capture long-range dependencies, which are crucial for tasks requiring global semantic understanding.

**Questions:**

Please see the weakness.

---

> ### Author Response · Authors · 2024-11-25
> **Response to 5BRv**
>
> Thank you for taking the time to read our paper and provide helpful feedback! We respond below to the points raised in your review.
>
> > I suggest the authors to conduct a thorough analysis of MaskEmbed's sensitivity to hyperparameters. This includes varying mask sizes, patch sampling strategies, and the influence of different reconstruction targets.
>
> We included a set of ablations that helped guide hyperparameter selection, please see Appendix B.1 in our updated submission. These include all the experiments you suggested, including sampling masks of varying sizes and with different patch sampling strategies (block-structured and random/unstructured), and the choice of reconstruction target. They also include ablations for the use of stronger augmentations, the training dataset and training duration, and we ran new loss function ablations requested by reviewer XvKA (discussed in their response). Finally, we also ran MaskEmbed with a variety of teacher models, and these results are shown in Figure 4 and Table 5 (including IN1k classifiers, CLIP, SigLIP, DFN, EVA02, MoCov3, etc).
>
> > Besides, including additional evaluations that specifically test the impact of MaskEmbed on global semantic understanding tasks may help to validate whether the method indeed compromises the model's ability to capture long-range dependencies.
>
> Thanks for this suggestion, it’s a good idea to include experiments for global semantic understanding. We’ve added new results where we fine-tune models for IN1k classification before and after locality alignment, and we verified that performance does not degrade but instead improves after locality alignment. This is likely because when learning patch-specific features, the model is also trained to retain its original prediction: the all-ones mask $m = 1$ corresponds to reconstructing the original teacher output $f(x)$, so preserving global understanding is built into our objective.
>
> Our IN1k fine-tuning results for CLIP models of different scales are shown in the following table, and we’ll include them in our revised submission. Our hyperparameters are chosen to match those in OpenCLIP. In addition to achieving better final results, locality-aligned models also trained significantly faster in the early epochs, suggesting that they learn strong features that are easy to adapt with fine-tuning.
>
> | Model              | Baseline | Aligned |
> | ---                | ---      | ---     |
> | CLIP ViT-B         | 82.4     | 82.9    |
> | CLIP ViT-L         | 84.4     | 85.4    |
> | CLIP ViT-L @ 336px | 85.0     | 85.5    |
>
> > This approach, while innovative, raises concerns about its impact on the encoder's ability to model global interactions and capture long-range dependencies, which are crucial for tasks requiring global semantic understanding.
>
> Thank you. Please note that in addition to our new IN1k results, our previous results also showed that locality alignment generally improves models’ global probing performance (Figure 4 and Table 5), and this improvement is consistent with our gains on VLM benchmarks (Figure 5). Overall, our results consistently show that loss of global semantic understanding is not an issue.

---

> > ### Author Response · Authors · 2024-12-02
> > **Discussion reminder**
> >
> > Thanks again for your helpful feedback. As we approach the end of the discussion phase, we hope you can read our rebuttal and let us know if your concerns have been resolved. We believe we've addressed your requests for ablations and evaluations of global semantic understanding, but we would be happy to discuss these points further.

---

### Author Response · Authors · 2024-11-25
**General response**

Thanks to all the reviewers for taking the time to read our work and provide helpful feedback. We’ve addressed all of your points in individual responses, and we’ll quickly summarize the main updates to our submission here.

- **New IN1k experiments:** to verify that locality alignment doesn’t hurt models’ global semantic understanding, we conducted new experiments fine-tuning CLIP backbones at multiple scales for IN1k classification before and after locality alignment. We found that performance does not degrade, but instead consistently improves after locality alignment
- **New dBOT comparison:** dBOT is a masked image modeling method with a bootstrapped random teacher, and our probing benchmark shows that it slightly outperforms MAE but is not competitive with CLIP. dBOT’s CLIP-teacher model yielded poor results, so we compared to EVA02 as a surrogate and found that our locality-aligned CLIP model has significantly better local and global probing performance
- **New loss function ablations:** we verified that reconstructing all the tokens is more effective than ignoring either the masked or unmasked tokens. We also showed that MSE slightly outperforms cosine similarity loss and L1 loss
- **Extended related work discussions:** to clarify differences with other methods combining distillation and masking, especially masked image modeling methods like MAE/dBOT/EVA02/BEiT, we added a new extended related work section in Appendix A
- **More information about VLM benchmarks:** we added additional information about our suite of benchmarks in Appendix D, and re-ran our VLM evaluations on the full version of each benchmark (our previous results were run on slim splits provided by Karamcheti et al., 2024)

---

### Author Response · Authors · 2024-11-27
**More on MIM comparisons (Part 1)**

Thanks again to all the reviewers for their helpful feedback. There were a few questions about masked image modeling (MIM) methods like dBOT/BEiT/MaskFeat/EVA02 and how they compare to locality alignment. Our previous response outlined key differences and showed favorable empirical comparisons, but we’d now like to supplement this with a better description of the relationship between these techniques. In short, **MIM is better viewed as an optional step before locality alignment than as a competing procedure.** They don’t do the same thing, but as we’ll show below they compose favorably when MIM is used before locality alignment.

To start, our reasoning about each procedure is the following:

- When you train a model $f$ with image-level supervision (e.g., CLIP), your representation $f(x)$ should reflect the global image label (e.g., “dog”). This is reflected in the final patch embeddings $f_1(x), \ldots, f_n(x)$, which don’t automatically capture patch-specific semantics (as demonstrated by our existing results)
- Next, when you train a MIM model $h$ with a pre-trained teacher $f$ (e.g., EVA02 and dBOT’s CLIP approach), your output for a masked image $m(x)$ should reflect a best guess for the patch features. Ideally, these should be $h_i(m(x)) = \mathbb{E}[f_i(x) \mid m(x)]$ (the expected embeddings conditioned on the available patches). After $h$ is trained, you can infer a lot by seeing how $h$’s outputs vary with the mask $m$. However, *this requires multiple masked queries* $h(m(x))$, and a single query with the full image $h(x)$ may not tell you much beyond an attempted reconstruction of the original patch embeddings $f(x)$
- Now, when we do locality alignment to get $g(x)$, the MIM model $h$ is exactly what we want as a teacher: it shows how the image semantics change due to masking, and locality alignment tries to compress all possible masked views $h(m(x))$ into its learned embeddings $g_1(x), \ldots, g_n(x)$. If these representations optimize our objective (eq. 1), then they can reconstruct each $h(m(x))$, and collectively contain sufficient information to tell differences due to each patch’s availability, and therefore determine local class contents

Our original submission actually explained this perspective. Our Section 3.3 about training data contains the following paragraph:

> Related to training data, we note that our approach only works as intended if the pre-trained model makes meaningful predictions with masked inputs. This can be ensured by pre-training with randomly dropped patches, which is performed for some but not all of the models in our experiments (He et al., 2022; Bao et al., 2021; Peng et al., 2022; Fang et al., 2024). Training or fine-tuning with random masking is often suggested in the interpretability literature [...] but we do not explore this direction and instead rely on the fact that ViTs empirically behave reasonably under masking (Naseer et al., 2021).

As mentioned in that snippet, we can get models that handle masking in two ways: 1) pre-train with random masking, or 2) pre-train without masking and then perform fine-tuning with masking (similar to MIM, but also explored by works like Frye et al., 2020; Jain et al., 2022). Option 1) seems more straightforward, but 2) is reasonable if we don’t want to pre-train from scratch. In our experiments, we did neither and relied on the fact that ViTs behave reasonably under masking, but this might have left performance on the table.

---

> ### Author Response · Authors · 2024-11-27
> **More on MIM comparisons (Part 2)**
>
> Now, we'll support this perspective with some experiments. Our arguments above suggest that MIM with a pre-trained teacher $f$ results in a model $h$ that’s perhaps a bit better than the original, but that locality alignment on the MIM version $h$ should lead to a much better model $g$. We conducted this comparison in our codebase with IN21k training data, $f$ as a CLIP ViT-B, and $h$ trained using MIM with a couple small improvements (fine-tuning vs training from scratch, using our $p(m)$ from Section 4.2, and using a loss that considers all patches). We then applied locality alignment to both the CLIP model $f$ and MIM model $h$, and and evaluated all the models on our probing benchmark. Here are the results:
>
> | Model                    | Teacher          | local | global |
> | ---                      | ---              | ---   | ---    |
> | CLIP ViT-B               | -                | 44.63 | 52.61  |
> | CLIP ViT-B (MIM)         | CLIP ViT-B       | 45.80 | 52.98  |
> | CLIP ViT-B (Aligned)     | CLIP ViT-B       | 46.32 | 54.55  |
> | CLIP ViT-B (MIM Aligned) | CLIP ViT-B (MIM) | 48.19 | 54.46  |
>
> The key observations are:
> 1. MIM improves upon the CLIP teacher, but not as much as locality alignment
> 2. **When we apply locality alignment to the MIM teacher, we get the best overall results:** the best local probing accuracy by a solid margin (consistent with our arguments above), and nearly the best global probing performance (roughly matching locality alignment applied directly to the CLIP teacher)
>
> To summarize, MIM seems less like an alternative to locality alignment than an ideal precursor. Doing this can add an extra step to our pipeline, unless the model is pre-trained with random masking; a simple alternative is to apply locality alignment directly, as we do in our experiments. If anything, this means we've understated the effectiveness of locality alignment in our current results.
>
> Overall, we hope this resolves the questions about the similarity between these ideas. They’re different procedures with different aims, but they’re complementary when applied together. We’ll be sure to clarify this in the revised paper, including a recommendation that researchers consider applying an initial MIM stage to maximize the benefits of locality alignment.

---

### Meta-Review · Area_Chair_HyN3 · 2024-12-19

**Metareview:**

This paper proposed a locality alignment method to improve the spatial understanding ability of visual encoders so as to improve the performance of vision-language models. Based on the insight that visual encoders pre-trained on text image pairs already contains local semantics, this paper proposes a simple fine-tune procedure named MaskEmbed to boost the spatial understanding ability of visual encoders. Experiments on several benchmark demonstrates the effectiveness of the proposed method.

All reviewers acknowledge that the proposed method is simple and effective. Most concerns proposed by reviewers (lack of comparison with existing methods, impact of fine-tuning on global semantics, ablation studies) are addressed in the discussion. One remaining concern is whether end-to-end fine-tuning could change the benefit of locality alignment. Considering the overall rating, I tend to accept this paper.

**Additional Comments On Reviewer Discussion:**

Reviewer 5BRv raised concerns about ablation studies on hyper parameters and the impact of fine-tuning on global semantics. The authors attempted to address these concerns in the discussion. As reviewer 5BRv did not response, from my judgement, these concerns are addressed.

Reviewer oNgu and XvKA acknowledged their concerns are addressed during the discussion and raised their score to accept (6 and 8).

After discussion, reviewer ARqQ still have concerns about the comparison with one-stage training method and whether end-to-end fine-tuning could change the benefit of locality alignment. As wether apply one-stage training or two stage-training is not the focus of this paper  and the authors justified the choice of one-state training, I feel this concern is not a big issue. However, the end-to-end fine-tuning concern is still valid and authors may want to better address this concern.

---

### Decision · Program_Chairs · 2025-01-22

Accept (Poster)